# Beyond Log Likelihood: Probability-Based Objectives for Supervised Fine-Tuning across the Model Capability Continuum

**Gaotang Li** [* 1]  **Ruizhong Qiu** [* 1]  **Xiusi Chen** [* 1]  **Heng Ji** [1]  **Hanghang Tong** [1]

## Abstract

Supervised fine-tuning (SFT) is the standard approach for post-training large language models (LLMs), yet it often shows limited generalization. We trace this limitation to its default training objective: negative log likelihood (NLL). While NLL is classically optimal when training from scratch, post-training operates in a different paradigm and could violate its optimality assumptions, where models already encode task-relevant priors and supervision can be long and noisy. In this work, we systematically study various probability-based objectives and characterize *when* and *why* different objectives succeed or fail under varying conditions. Through comprehensive experiments and extensive ablation studies across 8 model backbones, 27 benchmarks, and 7 domains, we uncover a critical dimension that governs objective behavior: the *model-capability continuum*. Near the *model-strong* end, prior-leaning objectives that downweight low-probability tokens (*e.g.*, $-p$, $-p^{10}$, thresholded variants) consistently outperform NLL; toward the *model-weak* end, NLL dominates; in between, no single objective prevails. Our theoretical analysis further elucidates how objectives trade places across the continuum, providing a principled foundation for adapting objectives to model capability. The code is available at https://github.com/GaotangLi/Beyond-Log-Likelihood.

## 1. Introduction

Supervised fine-tuning (SFT) has become a standard approach for post-training large language models (LLMs),

---
[*]Equal contribution [1]University of Illinois Urbana-Champaign. Correspondence to: Gaotang Li <gaotang3@illinois.edu>, Hanghang Tong <htong@illinois.edu>.

*Proceedings of the $43^{rd}$ International Conference on Machine Learning*, Seoul, South Korea. PMLR 306, 2026. Copyright 2026 by the author(s).

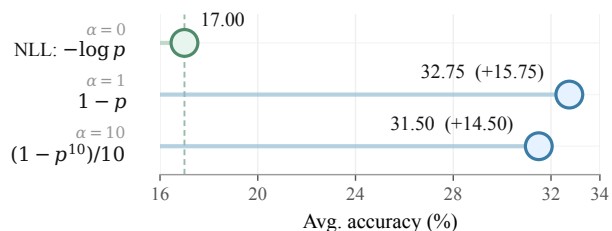

*Figure 1.* Motivating model-strong SFT result on math reasoning. For the objective family $f^\alpha(p) = (1 - p^\alpha)/\alpha$, where $\alpha \to 0$ recovers NLL ($-\log p$). Prior-leaning objectives with $\alpha = 1$ and $\alpha = 10$ substantially improve average accuracy over NLL.

widely used to elicit and strengthen their capabilities (Zhang et al., 2026b; Chung et al., 2024). Despite its popularity, many existing studies find that SFT often exhibits limited generalization (Ouyang et al., 2022; Chu et al., 2025). Nevertheless, this limitation may not arise from the imitation learning paradigm itself. Instead, we find that it may stem from its default training objective: negative log likelihood (NLL, $-\log p$). As a motivating case study, we generalize NLL into a parametrized family of learning objectives of the form $f^\alpha(p) := -\frac{p^\alpha-1}{\alpha}$, which includes NLL as a special case ($f^\alpha(p) \to -\log p$ as $\alpha \to 0$). We surprisingly find that other objectives significantly outperform NLL on some tasks, as shown in Fig. 1.

This unexpected observation motivates us to fundamentally revisit the training objective of SFT in its simplest form. While NLL has been shown to be optimal in classical learning theory when training from scratch on small-scale classification tasks (Cox, 1958; Zhang, 2004; Bartlett et al., 2006), LLM post-training operates in a fundamentally different paradigm that could degrade the optimality of NLL. Post-training begins with a pretrained model that already encodes task-relevant priors, and typically involves long chain-of-thought supervision spanning thousands of tokens that may be noisy. Requiring the pretrained model to replicate every token verbatim can hinder generalization.

To this end, we conduct a comprehensive study to demystify **which scenarios suit NLL and which suit other objectives**, *rather than advocating a single "one-size-fits-all" loss*. Our study uncovers a critical dimension that governs

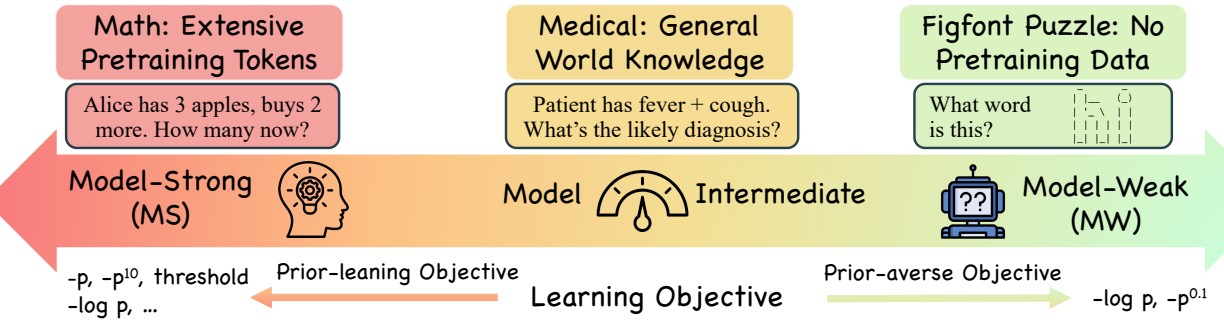

*Figure 2.* **The model capability continuum of SFT objectives in Post-Training.** At the model-strong (MS) end, where base models already encode extensive priors (e.g., Llama 3 reports 25% math pretraining tokens (Grattafiori et al., 2024)), prior-leaning objectives that downweight low-probability tokens (*e.g.*, $-p$, $-p^{10}$, or thresholded variants) consistently outperform NLL by up to 16%. At the model-weak (MW) end, where no useful priors exist (*e.g.*, no figfont puzzles in pretraining data), the standard NLL dominates. In the model-intermediate (MI) region (e.g., medical reasoning, where models rely on partial world knowledge), the gap between objectives narrows and no single choice consistently prevails. This continuum highlights how SFT objective effectiveness depends critically on base-model capability. Our results suggest that no single fixed SFT objective is universally optimal across settings.

the behavior of different objectives: the **model-capability continuum**. This continuum reflects the strength of prior signals inherited from pretraining: some domains (*e.g.*, math with abundant pretraining tokens) align well with the model's priors, while others (*e.g.*, novel puzzles with no pretraining exposure) do not, as illustrated in Fig. 2. Accordingly, the effectiveness of a learning objective depends on prior strength: prior-leaning objectives excel when priors are reliable, whereas prior-averse ones remain necessary when priors are weak.

We validate this perspective through extensive experiments spanning 8 model backbones, 27 benchmarks, and 7 domains. Our results reveal a clear continuum in how objectives behave: at the *model-strong* end, where base models already provide reliable priors, probability-based objectives that downweight low-probability tokens (e.g., $-p$, $-p^{10}$, or thresholded variants) consistently outperform NLL. At the *model-weak* end, where priors are misaligned with the data, NLL remains dominant by forcing the model to learn broadly from all tokens. In the intermediate region, the gap narrows and no single objective prevails. Further empirical analyses show that convexity and concavity of the learning objective, as a proxy for the degree to which model priors are respected, has opposite effects across the continuum. Likelihood estimation on the training set exhibits the same inversion.

To elucidate these findings, we provide theoretical underpinnings that characterize when and why different objectives outperform others. We characterize a sufficient condition showing that a more prior-leaning (*e.g.*, $-p$) achieve greater loss reduction than NLL in the model-strong end in gradient flow. The opposite holds in the model-weak end, where NLL achieves larger reductions. This theoretical characterization mirrors our empirical results and provides a principled explanation of how the objective form

and the model capability interact.

## 2. Related Works

**Improving SFT from RL perspective.** Motivated by the success of reinforcement learning in reasoning tasks, a growing body of work reinterprets and improves SFT through an RL lens. Wang et al. (2026) cast SFT and DPO as instances of implicit reward learning, suggesting that smaller learning rates and alternative divergence-based objectives can improve performance. Qin & Springenberg (2025) further integrate importance sampling into SFT, while Zhu et al. (2026) introduces a PPO-style clipped to constrain policy drift. Closest to our setting, Wu et al. (2026) propose uniformly reweighting gradient coefficients, which is essentially equivalent to our $-p$ objective. Collectively, these RL-inspired approaches can be interpreted as implementing *prior-leaning* updates within our framework. While prior work demonstrates their effectiveness in certain domains, our model-capability continuum view shows that such updates can fail in others, highlighting that no single loss is universally optimal across settings.

**Other SFT loss functions.** Beyond RL-motivated objectives, a number of alternative losses to NLL have been explored in supervised learning, including mean squared error, focal loss (Lin et al., 2017), and Huber loss (Huber, 1992). These *distribution-based* losses can be naturally understood through our framework. More recent proposals, such as entropic distribution matching (Li et al., 2025), introduce additional regularization terms and can also be interpreted within our framework as *prior-leaning* objectives. Instead of adding objective sophistication or targeting specific applications, our work uncovers a model-capability continuum through simple probability-based objectives across diverse settings. Additional discussion of

related loss functions is provided in Appen. F.2, with a full version of the related work deferred to Appen. A.

**Positioning of our work.** We provide the *first* capability-based characterization of SFT objectives in LLM post-training. Instead of promoting a single "best" loss, we establish a principled account of when and why objectives trade advantages across the model-capability continuum.

## 3. A Unified Categorization of SFT Objectives

**Language Model Post-Training.** We focus on the post-training stage of large language models (LLMs). Let $p_\theta$ denote a pretrained base model that has already undergone large-scale pretraining and accumulated extensive world knowledge. Such models typically produce predictions that are reasonably well-calibrated (Zhu et al., 2023; Xie et al., 2024), and their outputs encode task-relevant priors derived from pretraining corpora.

**Standard Supervised Fine-Tuning.** We consider supervised fine-tuning (SFT) on a dataset $T$ of input-output pairs $(x, \tilde{y})$, where $\tilde{y} = (y_1, \ldots, y_N)$ denotes the target sequence. The model defines token-level conditionals $p_\theta(y_t \mid y_{<t}, x)$. At decoding step $t$, let $z_t \in \mathbb{R}^V$ denote the logits over the vocabulary, $p_t = \mathrm{softmax}(z_t)$, and $p_{t,i} = \mathrm{softmax}(z_t)_i$. For brevity, write $y = y_t$, and denote by $\delta_{i,y}$ the Kronecker delta. In standard SFT, the training objective is to minimize the negative log likelihood, equivalently the cross-entropy loss, over the dataset:

$$\mathcal{L}_{\log(p)}(\theta) = \mathbb{E}_{(x,\tilde{y})\sim T}\big[-\log p_\theta(\tilde{y} \mid x)\big]$$
$$= \mathbb{E}_{(x,\tilde{y})\sim T}\left[\sum_{t=1}^{N} -\log p_\theta(y_t \mid y_{<t}, x)\right]. \quad (1)$$

**A General Family of Probability-Based Objectives.** We now extend beyond log likelihood by considering a broader family of objectives. For any differentiable and nonincreasing function $f : [0, 1] \to \mathbb{R}$, we define

$$\mathcal{L}_{f(p)}(\theta) = \mathbb{E}_{(x,\tilde{y})\sim T}\big[f\big(p_\theta(\tilde{y} \mid x)\big)\big]$$
$$= \mathbb{E}_{(x,\tilde{y})\sim T}\left[\sum_{t=1}^{N} f\big(p_\theta(y_t \mid y_{<t}, x)\big)\right]. \quad (2)$$

One useful general instance of $f$ is given by

$$f^\alpha(p) = \frac{1 - p^\alpha}{\alpha}. \quad (3)$$

As $\alpha \to 0$, it reduces to $f^\alpha(p) \to -\log(p)$ (NLL). When $\alpha = 1$, it yields the plain-$p$ objective $f^\alpha(p) = 1 - p$, which corresponds to *maximizing the expected average prediction accuracy*. More generally, the function is concave when $\alpha \geq 1$ and convex when $0 \leq \alpha \leq 1$.

**Prior-learning versus Prior-averse Objectives.** The key distinction among these objectives lies in the form of their gradients with respect to the *correct logit class*, which governs the resulting learning dynamics.

**Lemma 3.1** (Gradient Shape). *Let $f : [0,1] \to \mathbb{R}$ be differentiable and nonincreasing. Then the gradient of Eq. with respect to the logits at step $t$ is*

$$\frac{\partial(\mathcal{L}_f)}{\partial z_{t,i}} = s_f(p_{t,y})\,(\delta_{i,y} - p_{t,i}),$$

*where $s_f(p) \triangleq -f'(p)\,p \geq 0$, $\delta_{iy} = \mathbf{1}\{i = y\}$.*

*In particular, for the correct class $i = y$,*

$$\frac{\partial(\mathcal{L}_f)}{\partial z_{t,y}} = s_f(p_{t,y})\,(1 - p_{t,y}) = W_f(p_{t,y}),$$
$$W_f(p) \triangleq -f'(p)\,p\,(1 - p).$$

**Proposition 3.2** (Convex versus Concave Objectives). *Let $f \in C^2[0,1]$ with $f'(p) < 0$ for all $p \in (0,1)$. Define $W_f(p) = -f'(p)\,p(1-p)$. Then if $f$ is concave, any maximizer of $W_f$ lies in the interval $[\frac{1}{2}, 1]$; if $f$ is convex, any maximizer of $W_f$ lies in the interval $[0, \frac{1}{2}]$.*

*In other words, convex objectives emphasize gradient contributions from low-probability tokens, while concave objectives emphasize high-probability tokens.*

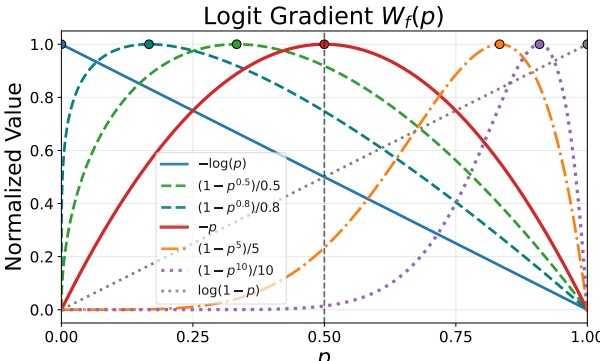

*Figure 3.* Logit-gradient weights $W_f(p)$ for representative objectives. The marked peak of each curve shows the token-probability region where the correct-logit gradient is largest, distinguishing prior-averse objectives from prior-leaning objectives.

The weighting term $W_f(p)$ determines how much learning signal each token contributes relative to the model's prior belief. For the parametric family in Eq. 3, we have $W_f(p) = p^\alpha(1 - p)$. As $\alpha \to 0$ (NLL), this reduces to $W_f(p) \to (1 - p)$, which strongly emphasizes low-probability tokens. When $\alpha \geq 1$ ($f(p) = 1 - p$), the gradient signal from low-probability tokens quickly diminishes. For a special case $f(p) = -\log(1-p)$, we obtain $W_f(p) =$

$p$, which exhibits the opposite trend of $-\log(p)$ by emphasizing high-probability tokens. Fig. 3 visualizes these gradient shapes $W_f(p)$ for different objectives: the dot marks where each logit-gradient weight is largest, and the dashed line at $p = 0.5$ serves as a reference point separating objectives that favor low- versus high-probability tokens. More formally, Prop. 3.2 shows that convex objectives (e.g., $-\log p$) achieve their maximum within $[0, 0.5]$, thus prioritizing low-probability tokens (*prior-averse*); whereas concave objectives (e.g., $-p^2$) peak within $[0.5, 1]$, thereby reinforcing already confident predictions (*prior-leaning*). This distinction illustrates how convexity modulates the degree to which an objective respects model priors. In particular, the family in Eq. 3 can be seen as providing a smooth transition between prior-averse and prior-leaning behavior. This leads to the following definition.

**Definition 3.3** (Prior-leaning versus Prior-adverse Objectives). We classify objectives according to how $W_f$ distributes its mass over $p$. We say the objective is:

- *Prior-leaning* if the majority of gradient weight is concentrated on medium- to high-probability tokens (i.e., $p$ above a threshold $\tau$), thereby leveraging the model's prior to refine already plausible predictions.

- *Prior-averse* if the majority of gradient weight is concentrated on low-probability tokens ($p$ below $\tau$), pushing the model to learn from unlikely predictions.

This definition emphasizes that different objectives exploit the model's prior in opposite ways. While the precise boundary between prior-leaning and prior-averse (e.g., the choice of threshold $\tau$) is not unique and may depend on the task, some objectives exhibit clear contrasts (*e.g.*, $-\log p$ versus $-p$), which form the primary focus of our study. To further probe their behavior, we also consider a hard-thresholding variant:

$$\mathcal{L}_{\mathrm{HT}(I), f(p)}(\theta) = \mathbb{E}_{(x,\tilde{y}) \sim T}\left[ f\left(p(\tilde{y} \mid x)\right) \mathbf{1}\{p(\tilde{y} \mid x) \in I\}\right],$$
$$(4)$$

where $\mathrm{HT}(I)$ denotes restricting updates to tokens whose predicted probabilities fall within an interval $I \subseteq [0, 1]$. This formulation is very useful for ablation, as it isolates the contribution of tokens in specific probability ranges.

**The model capability continuum.** Unlike traditional classification tasks, language model post-training spans a wide variety of domains that differ substantially in how well they are supported by pretraining. Consequently, not all tasks should be treated uniformly. We categorize tasks along a *model-capability continuum*, defined by the strength of the base model prior. A general categorization is shown in Fig. 2. Our classification relies on two complementary perspectives: (1) From the pretraining data side, tasks differ in

the portion of relevant data contained in the corpus. For example, the LLaMA-3 report indicates that $\sim$25% of its pretraining tokens are math-related, suggesting strong priors for mathematical reasoning (*model-strong*). By contrast, figfont puzzles fall entirely outside the pretraining corpus and thus represent *model-weak* tasks, while domains with partial coverage, such as medical reasoning, are considered *intermediate*. (2) From the model side, we use the mean predicted probability on the training set as a quantitative proxy for prior strength. This is analogous to how the widely-used benchmark lm_eval evaluates base models by computing log-likelihood over answer tokens across many benchmarks (Gao et al., 2023). This measure aligns well with intuition: math tasks achieve high predicted likelihood of the training even before SFT (e.g., Qwen2.5-Math-7B: 0.81, LLaMA-3.1-8B: 0.76), whereas medical reasoning lies in the middle ($\sim$0.50), and figfont puzzles remain extremely low ($\sim$0.01). Together, these perspectives motivate our continuum view and ground it in both qualitative and quantitative evidence. The details and the rationales about our classification are included in Appen. B.1.

At the model strong (MS) end, prior-leaning objectives can be leveraged to refine a small number of critical tokens by concentrating learning on mid- to high-probability tokens that are more likely to be correct. At the model weak (MW) end, prior-averse objectives are more suitable, as they encourage the model to improve predictions across all tokens. For models of intermediate capability (MI), both objectives may provide benefits, depending on the characteristics of the task and the base model.

## 4. Main Experiments

In this section, we empirically validate the proposed continuum view of SFT post-training and evaluate the performance of different probability-based objective functions.

### 4.1. Experimental Setup

To empirically validate the continuum view, we conduct experiments across three representative domains: mathematical reasoning, medical reasoning, and textual puzzles, which serve as anchor settings for the model-capability continuum. As motivated in Sec. 3, these domains occupy different positions along the model-capability continuum. For the *model-strong (MS)* end, we use NuminaMath (LI et al., 2024) as training data. For the *model-weak (MW)* end, we generate synthetic figfont puzzles from Reasoning Gym (Stojanovski et al., 2025). For the *intermediate (MI)* region, we adopt m23k (Huang et al., 2026), a high-quality medical reasoning dataset. Additional statistics supporting this classification are provided in Appen. B.1. To test whether the same continuum behavior extends beyond these anchor domains, we also include fixed-domain ca-

*Table 1.* Main results in the Model Strong (MS) end. Both prior-leaning objectives $-p$ and thresholded $-\log(p)$ consistently outperform the prior-averse $-\log(p)$ objective across models and datasets. Best results are in bold.

| Models | Math500 | Minerva Math | Olympiad Bench | AIME24 | AMC23 | Avg. |
|---|---|---|---|---|---|---|
| **LLaMA-3.1-8B** | | | | | | |
| Base | 1.76 | 0.68 | 0.86 | 0.00 | 1.25 | 0.91 |
| -log(p) | 17.59 | 5.84 | 3.04 | 0.21 | 5.78 | 6.49 |
| -log(p)$\mathbf{1}\{p \geq 0.2\}$ | 24.39 | **10.49** | 5.10 | **0.41** | **11.25** | 10.33 |
| -p | **25.29** | 10.09 | **6.37** | **0.41** | 10.62 | **10.56** |
| **DeepSeekMath-7B** | | | | | | |
| Base | 5.70 | 2.89 | 1.51 | 0.00 | 2.34 | 2.49 |
| -log(p) | 28.79 | 9.29 | 6.57 | 0.21 | 10.62 | 11.10 |
| -log(p)$\mathbf{1}\{p \geq 0.2\}$ | **40.38** | 19.38 | **13.98** | 0.62 | 18.91 | 18.65 |
| -p | 39.55 | **20.14** | **13.99** | **1.24** | **20.62** | **19.11** |
| **Qwen2.5-Math-1.5B** | | | | | | |
| Base | 30.71 | 8.81 | 14.88 | 2.49 | 17.97 | 14.97 |
| -log(p) | 42.52 | 12.71 | 12.09 | 0.62 | 17.03 | 17.00 |
| -log(p)$\mathbf{1}\{p \geq 0.2\}$ | 63.95 | 24.79 | 26.08 | **7.09** | **38.28** | 32.04 |
| -p | **65.27** | **26.18** | **26.66** | 6.88 | 38.13 | **32.75** |
| **Qwen2.5-Math-7B** | | | | | | |
| Base | 40.38 | 13.66 | 16.36 | 6.04 | 24.69 | 20.23 |
| -log(p) | 51.90 | 18.88 | 17.37 | 2.70 | 22.50 | 22.67 |
| -log(p)$\mathbf{1}\{p \geq 0.2\}$ | 67.85 | **32.47** | **33.90** | **8.76** | **47.81** | **38.16** |
| -p | **68.47** | 31.99 | 32.26 | 8.75 | 41.09 | 36.51 |

pability sweeps in general instruction tuning (Sec. 4.3 and Appen. C.2) and math (Appen. C.3), as well as additional domains that instantiate the MS and MW ends: coding and low-resource multilingual instruction tuning (Appen. C.3, C.4).

Across the main and appendix studies, our experiments cover a diverse set of advanced backbones, including LLaMA-3.1-3B, LLaMA-3.2-3B, LLaMA-3.1-8B, DeepSeekMath-7B, Qwen2.5-Math-1.5B, Qwen2.5-Math-7B, Qwen2.5-1.5B/3B/7B/14B/32B, and Qwen2.5-Coder-7B. We primarily compare the $-p$ and $-\log p$ objectives, with one exception: on the MS end, we also evaluate a thresholded variant of $-\log p$ that excludes low-probability tokens. The evaluation datasets, training details, and further experimental configurations are provided in Appen. B.

### 4.2. Main Results

**Model-Strong Results Interpretation.** Tab. 1 reports results in the model-strong (MS) end, where base models already exhibit strong priors aligned with the ground truth. In this setting, the $-p$ objective consistently outperforms standard negative log-likelihood ($-\log p$). This trend suggests that when model predictions are already reliable, a prior-leaning objective like $-p$ better capitalizes on high-confidence tokens by suppressing the influence of low-probability ones. To further dissect this effect, we evaluate a thresholded variant of $-\log p$ that excludes tokens with $p < 0.2$. This adjustment directly mitigates the effect

of low-confidence tokens and leads to consistent improvements over standard $-\log p$. In many cases, it performs on par with, or even surpasses, $-p$ applied to full tokens. Such evidence highlights that the weakness of standard NLL in this setting lies in its excessive emphasis on low-probability tokens. Prior-leaning objectives that explicitly downweight low-confidence tokens consistently yield the greatest gains at the MS end. This MS behavior is not restricted to the math domain: the coding experiments in Appen. C.3 show the same preference for $-p$ over $-\log p$, and the fixed-domain math scaling study further shows that the advantage of $-p$ increases as the backbone becomes stronger. We provide further empirical analysis in Sec. 5 with a more careful study of the pattern.

**Model-Intermediate Results Interpretation.** In Tab. 2, results on medical reasoning reveal a strikingly different pattern: the performance of $-p$ and $-\log p$ is nearly indistinguishable, with differences well within statistical variation. This neutrality arises from the nature of intermediate priors. On one hand, the priors are not strong enough for the prior-leaning objective $-p$ to yield consistent refinements; on the other, they are not weak enough for the prior-averse objective $-\log p$ to offer a decisive corrective advantage. This observation is important because it indicates the existence of a region where gains are unlikely to come from altering the learning objective itself. Instead, improvements may rely on alternative directions, such as better data curation or targeted domain supervision.

*Table 2.* Main results in the Model Intermediate (MI) region. Both $-p$ and $-\log(p)$ result in similar performance.

| Model | MedMC | MedQA | PubMed | MMLU-P | GPQA | Lancet | MedB (4) | MedB (5) | MedX | NEJM | Avg. |
|---|---|---|---|---|---|---|---|---|---|---|---|
| | | | | | LLaMA-3.1-3B | | | | | | |
| Base | 21.30 | 21.92 | 22.60 | 11.40 | 23.08 | 25.00 | 23.05 | 15.26 | 10.35 | 23.22 | 19.48 |
| -log(p) | **42.60** | **45.56** | **67.40** | **38.63** | 24.36 | **46.84** | **46.10** | **34.42** | 11.59 | **43.28** | **37.99** |
| -p | 39.42 | 41.95 | 62.70 | 33.88 | **38.46** | 44.17 | 35.71 | 28.57 | **12.63** | 40.80 | 36.29 |
| | | | | | LLaMA-3.1-8B | | | | | | |
| Base | 23.57 | 29.14 | 21.00 | 20.00 | 29.49 | 22.57 | 30.52 | 20.45 | 10.01 | 20.73 | 21.89 |
| -log(p) | **55.08** | **59.47** | 74.00 | **53.62** | 32.05 | **57.28** | **52.27** | **46.10** | **15.87** | **59.20** | **47.23** |
| -p | 54.10 | 58.44 | **76.50** | 52.70 | **44.87** | 54.13 | 42.21 | 42.53 | 13.80 | 54.73 | 45.89 |
| | | | | | Qwen2.5-1.5B | | | | | | |
| Base | 22.21 | 21.84 | 18.50 | 11.21 | 24.36 | 22.57 | 24.03 | 17.53 | 10.84 | 18.74 | 18.59 |
| -log(p) | **39.64** | **39.59** | 66.70 | 34.92 | 33.33 | **38.83** | **38.31** | 27.60 | 10.56 | 34.16 | **35.13** |
| -p | 38.58 | 36.68 | **68.00** | **38.37** | **35.90** | 35.68 | 36.69 | **28.90** | **11.94** | **39.97** | 35.02 |
| | | | | | Qwen2.5-Math-7B | | | | | | |
| Base | 35.84 | 27.26 | 49.30 | 30.23 | 35.90 | 30.34 | 24.03 | 18.18 | 10.21 | 24.71 | 27.55 |
| -log(p) | **36.48** | 33.78 | **72.60** | 35.50 | 38.46 | **40.05** | **29.87** | 26.95 | **10.42** | 26.70 | 33.56 |
| -p | 35.62 | 33.78 | 69.90 | **38.83** | **42.31** | 35.44 | 33.12 | 27.60 | 10.49 | 26.70 | **33.83** |

*Table 3.* Main results in the Model Weak (MW) end. $-\log(p)$ consistently outperforms $-p$. Best results are in bold.

| | LLaMA-3.2-3B | | | LLaMA-3.1-8B | | | Qwen2.5-1.5B | | | Qwen2.5-7B | | |
|---|---|---|---|---|---|---|---|---|---|---|---|---|
| Metric | Base | -log(p) | -p | Base | -log(p) | -p | Base | -log(p) | -p | Base | -log(p) | -p |
| Exact Match | 0.00 | **1.08** | 0.00 | 0.00 | **1.34** | 0.00 | 0.00 | **0.60** | 0.0 | 0.00 | **35.20** | 0.00 |
| Jaro-Winkler Similarity | 41.89 | **44.39** | 2.43 | 30.17 | **43.59** | 10.15 | 35.32 | **32.98** | 8.36 | 44.92 | **82.48** | 10.15 |

**Model-Weak Results Interpretation.** Tab. 3 reveals the opposite trend at the MW end: $-\log p$ consistently outperforms $-p$, often by substantial margins. When priors are poorly aligned with the ground truth, the prior-leaning objective $-p$ allocates disproportionate weight to unreliable high-probability tokens, thereby reinforcing errors. By contrast, the prior-averse $-\log p$ ensures that low-probability tokens, which often correspond to mistakes, receive stronger gradient signals, forcing the model to correct its errors and spread learning more broadly across the output distribution. This explains why NLL, despite its shortcomings elsewhere, remains the most effective objective in weak-prior settings. This MW pattern is also reproduced outside figfont puzzles: in low-resource multilingual instruction tuning, $-\log p$ consistently outperforms $-p$ for both LLaMA-3.1-8B and Qwen2.5-7B (Appen. C.4). Consequently, progress on MW tasks is more likely to come from stronger or more targeted supervision or other methods of injecting knowledge. We provide further empirical analysis in Sec. 5 with a more careful study of the pattern.

### 4.3. Experiments on General Instruction Tuning

To further demonstrate the breadth of our continuum view, we run additional general instruction-tuning experiments comparing the two objectives, $-p$ and $-\log p$. Experimental details and the full results are deferred to Appen. C.2. We fine-tune three Qwen2.5 base models spanning 3B to 14B parameters, and evaluate the resulting models trained under each objective. Figure 4 reports the head-to-head win rate between the two final models at each scale. The results mirror the same continuum behavior observed elsewhere: as model scale increases, the preferred objective shifts across the MS, MI, and MW regimes, supporting the generality of our claim beyond previous settings.

**Fixed-domain capability sweep.** Here the data mixture, evaluation protocol, and objective comparison are held fixed, while only the base-model scale changes. The smooth transition from an NLL-favoring smaller model to a $-p$-favoring larger model therefore shows that the continuum behavior also appears within a single broad instruction-tuning setting.

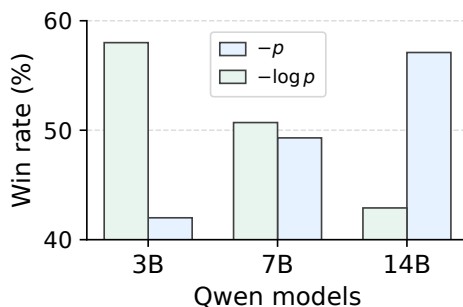

*Figure 4.* Head-to-head win rate of the $-p$-trained model against the $-\log(p)$-trained model on AlpacaEval2. As backbone size increases from 3B to 14B, the prior-leaning objective becomes more effective.

## 4.4. Additional Experiments

**Generalization across scales and domains.** The appendix experiments further show that the observed reversal is not limited to the three anchor domains. On the fixed math domain, Appen. C.3 shows that the performance gap between $-p$ and $-\log p$ widens as the Qwen2.5 backbone scales from 3B to 32B. Across domains, it also reports coding results where $-p$ again outperforms $-\log p$, while Appen. C.4 reports low-resource multilingual instruction tuning results where $-\log p$ consistently outperforms $-p$. Together with the main experiments above, these results strongly support a capability-continuum interpretation.

## 5. Empirical Analysis

In this section, we provide a deeper empirical analysis of the findings in Sec. 4, with a particular emphasis on the MS and MW ends where the choice of training objective has the largest effect. Our goal is to move beyond merely reporting performance numbers and to analyze the mechanisms that drive the observed differences. To this end, we structure the analysis around three guiding questions:

1. In the MS end, what mechanisms explain the underperformance of NLL?
2. How do objectives with different emphasis on model priors behave across the two ends?
3. To what extent are these objectives consistent with likelihood estimation on the training set?

Answering these questions provides a deeper understanding of how different objectives interact with model capability from complementary perspectives.

**Model Setup.** For ablation studies in the MS end, we focus on Qwen-2.5-Math-1.5B, which shows the clearest gap between objectives. For the MW end, we use Qwen-2.5-7B. All training details and evaluation protocols remain identical to those in Sec. 4, ensuring that differences arise solely from the choice of objective.

## 5.1. Ablation on Quantile Thresholding with Different Objectives

**Detailed Setup.** This ablation examines how restricting training to different quantiles of tokens affects the relative performance of objectives. We compare three instances of $f(p)$ in Eq. : $-\log(p)$, $-p$, and $\log(1-p)$, which emphasize low-, mid-, and high-probability tokens, respectively (shown in Fig. 3). All experiments are identical except for the subset of tokens selected by the quantile thresholding rule in Eq. 4. Quantile thresholds are computed from the base model's predicted token probabilities prior to training. We apply both bottom thresholding and top thresholding, denoted by ($\geq$ Percentile) and ($\leq$ Percentile), respectively.

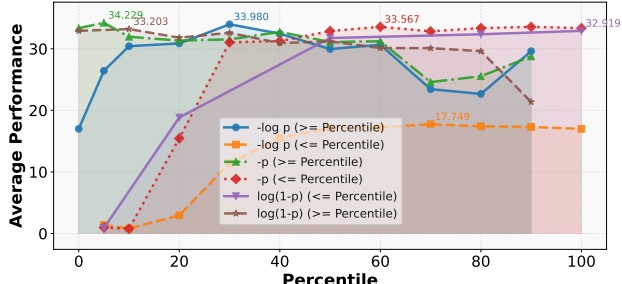

*Figure 5.* **Performance under quantile thresholding** for $-\log(p)$, $-p$, and $\log(1-p)$. Let $Q_{\text{percentile}}$ denote the predicted probability at the specified percentile of the training set. ($\geq$ Percentile) corresponds to $I = [Q_{\text{percentile}}, 1]$ in Eq. 4, while ($\leq$ Percentile) corresponds to $I = [0, Q_{\text{percentile}}]$. Key findings: (1) low-probability tokens consistently harm performance across all objectives; (2) when training on all tokens, objectives that de-emphasize low-probability tokens ($-p$ and $\log(1-p)$) outperform $-\log(p)$; (3) restricting training to only the top 10% of tokens yields the strongest improvements across all objectives, surpassing standard SFT.

Bottom thresholds vary from 5% to 100%, and top thresholds vary from 0% to 90%.

**Results Interpretation.** The results in Fig. 5 reveal consistent patterns that align with our main experiments in Sec. 4. First, all objectives achieve strong performance when restricted to only the top 10% tokens, significantly exceeding standard NLL on all tokens. Second, performance drops sharply when training on low-probability tokens, confirming that they contribute adversarially to learning. Third, when applying bottom-thresholding, $-p$ and $\log(1-p)$ consistently outperform $-\log(p)$, illustrating the benefits of objectives that de-emphasize unreliable tokens. Finally, the degradation of $\log(p)$ performance when trained on all tokens (blue curve) can be largely attributed to the bottom 10% quantile. Overall, these results reinforce the main conclusion from Sec. 4: in the MS end, *low-probability tokens act primarily as noise to the strong model.*

## 5.2. Objective Convexity and Performance Difference

**Detailed Setup.** To systematically examine the effect of objective on downstream performance, we study the parametric family in Eq. 3. This objective is concave when $\alpha \geq 1$ and convex when $\alpha \leq 1$. A "more concave" objective is more prior-leaning and vice versa, as shown in Fig. 3. We leverage the convexity of this objective as a proxy for assessing prior-leaning versus prior-averse objectives. We vary $\alpha$ from 0.1 to 1.0 in increments of 0.1, and from 1.0 to 10.0 in increments of 1.0.

**Results Interpretation.** As shown in Fig. 6, convexity affects performance in opposite directions across the SFT continuum. In the MS end, accuracy improves as $\alpha$ in-

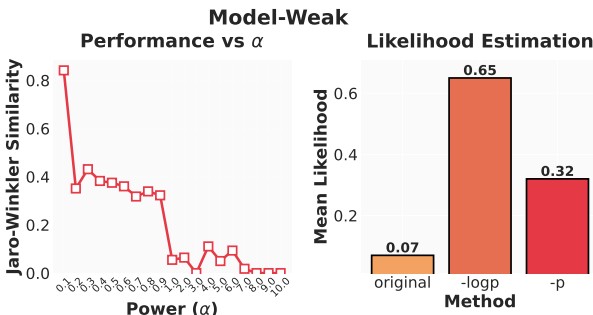

*Figure 6.* Analysis of MS and MW ends in terms of objective convexity (with Eq. 3) and likelihood estimation. In MS, more concave (prior-leaning) objectives yield better downstream accuracy, while in MW, more convex (prior-averse) objectives dominate. The likelihood estimation results align with these trends, suggesting that objective shape directly interacts with model prior strength.

creases, peaking near $\alpha = 1$ and remaining stable for larger values. In the MW end, performance is maximized at $\alpha = 0.1$ and deteriorates rapidly as $\alpha$ approaches 1 and exceeds the convexity boundary. This dichotomy highlights the importance of aligning objective shape with model prior strength: concave objectives (that emphasize model priors) are more effective when priors are strong, while convex objectives (that de-emphasize model priors) are preferable when priors are weak.

### 5.3. Likelihood Estimation on the Training Set

**Detailed Setup.** In this ablation, we evaluate the empirical training performance of different objectives by computing the average predicted likelihood on the training set before and after fine-tuning:

$$\text{Likelihood Estimation}(\theta) := \frac{1}{N} \sum_{i=1}^{n} \sum_{j=1}^{|\tilde{y}_i|} p_\theta(\tilde{y}_{i,j}). \quad (5)$$

where $i$ denotes the $i$-th sample and $j$ denotes the $j$-th token, and $N = \sum_{i=1}^{n} |\tilde{y}_i|$, the total number of training tokens. We focus on comparing $-p$ and $-\log(p)$ in both the MS and MW ends.

**Results Interpretation.** The likelihood estimation results, shown in Fig. 6, closely parallel the downstream accuracy trends. In the MS end, $-p$ achieves higher mean predicted probabilities, confirming that they better align with strong model priors and effectively capture the training distribution. In contrast, in the MW end, $-\log(p)$ yield higher training performance, reflecting their ability to correct misaligned priors by emphasizing low-probability tokens. These findings indicate that the interaction between objective shape and regime governs not only generalization performance but also the model's fit to the training data.

## 6. Theoretical Analysis

### 6.1. Setup

**Data.** Let the input prompt be $x \in \mathcal{X}$. The *true* conditional distribution over tokens $y \in [V]$ is denoted by $r(y \mid x)$, with $y^* \sim r(\cdot \mid x)$. We write $\mathcal{D}$ for the marginal distribution over pairs $(x, r(\cdot \mid x))$, and let $T(\cdot \mid x)$ denote the empirical training distribution over contexts $x$, which we abuse the notation for writing $(x, \tilde{y}) \sim T$. We use subscript $p_{(\cdot)}$ to denote model predictions $p(\cdot)$.

**Model and objectives.** Let $p_\theta(\cdot \mid x) = \text{softmax}(z_\theta(x))$ be the next-token distribution of an autoregressive LM with parameters $\theta$, and write $p_0(\cdot \mid x) = p_{\theta_0}(\cdot \mid x)$ for the base model. We define the *population risk* to be

$$\mathcal{R}(\theta) = \mathbb{E}_{(x,y^*) \sim \mathcal{D}, y^p \sim p_\theta(\cdot|x)} \left[ -\mathbb{1}\{y^* = y^p\} \right],$$

During SFT we minimize the empirical objective

$$\mathcal{L}_f(\theta) = \mathbb{E}_{(x,\tilde{y}) \sim T} \left[ f\big(p_\theta(\tilde{y} \mid x)\big) \right]$$

where $f : [0,1] \to \mathbb{R}$ is differentiable and decreasing in $p$. Our theoretical analysis mainly relies on the following assumption about the two ends of the continuum:

**Assumption 6.1** (Model-Capability Assumption). We make the following assumptions about model capability in the Model-Strong and Model-Weak ends:

- **Model-Weak.** In the MW end, we assume that model predictions are uniform over the vocabulary $V$, where $2/V < 0.55$.

- **Model-Strong.** In the MS end, we assume that for any given $x$, $\Pr_{y^*, \tilde{y}} [(p_{y^*} + p_{\tilde{y}}) \geq 0.55] \geq K$ with $K \geq 0.70$.

**Assumption 6.2** (Trainable Base Model). We assume that the base model is still not perfect: for any given $x$, $\Pr[0.55 \leq (p_{y^*} + p_{\tilde{y}}) \leq 0.95] \geq 1 - K$ in the MS end.

*Remark* 6.3. The MW assumption captures the essential condition of weakness by modeling the base as uninformative. The MS assumption is grounded in practice: in Appen. C.1, we empirically validate this. For a uniform predictor, $p_{y^*} + p_{\tilde{y}} = 2/V$, so $2/V < 0.55$ ensures that the MW assumption does not satisfy the MS threshold. This condition is trivially true for LLM-scale vocabularies. Assumption 6.2 is mild and simply guarantees that optimization is nontrivial. We choose $1 - K$ for simplicity of proof.

### 6.2. Main Results

We analyze the optimization dynamics of different objectives under gradient flow. For an objective $f_i$, let $\dot{\theta}_t^{(i)} = -\nabla \mathcal{L}_{f_i}(\theta)$ denote the corresponding gradient flow, and let $\mathcal{R}(\theta_t^{(i)})$ be the population risk at time $t$. Our goal is to maximize the reduction in risk, as captured by $\dot{\mathcal{R}}(\theta_t^{(i)})$.

**Theorem 6.4** (Characterization via Gradient Flow, Informal). *Suppose that $f_2'(p) - f_1'(p) < 0$ for all $\tilde{p}$, and Assumptions 6.1–6.2 hold. Then, in a simplified setup, we have the following conclusions:*

- $\dot{\mathcal{R}}(\theta_t^{(1)})\big|_{t=0} \geq \dot{\mathcal{R}}(\theta_t^{(2)})\big|_{t=0}$ *in Model Strong End.*

- $\dot{\mathcal{R}}(\theta_t^{(1)})\big|_{t=0} \leq \dot{\mathcal{R}}(\theta_t^{(2)})\big|_{t=0}$ *in Model Weak End.*

*Remark* 6.5. This theorem characterizes a sufficient condition for which the relative advantage of two objectives reverses across the MS and MW ends. For example, by setting $f_1(p) = 1 - p$ and $f_2(q) = -\log p$, we can conclude that in the model-strong end, the prior-leaning $-p$ objective achieves larger risk reduction than NLL, whereas in the model-weak end, NLL is superior. This reversal mirrors our empirical observations and highlights the central theme of this work: the effectiveness of an SFT objective depends critically on model capability. The full version of the theoretical analysis is available at Appen. E.

## 7. Discussion

In this work, we revisited supervised fine-tuning (SFT) objectives for large language model post-training and showed that negative log likelihood (NLL), while classically optimal from scratch, is not universally effective once models already encode priors and supervision is long and noisy. Our central contribution is the *model-capability continuum*, instantiated through a general family of probability-based objectives, which shows that objective effectiveness depends critically on the prior strength of the base model. Across extensive analyses, we find that objectives reverse their relative advantage across different regions, yielding a unified explanation of how objective form interacts with model capability. Appen. F further situates several well-known existing objectives in this lens: Focal loss is more

prior-averse than NLL, Huber-style probability losses are prior-leaning, and KL-regularized or RL-style objectives favor high-probability behaviors.

For practitioners, the continuum gives a simple diagnostic for objective choice. If the target task is well represented in pretraining, prior-leaning objectives such as $-p$ or thresholded NLL are natural candidates. If the task is poorly covered and predictions are close to uninformative, NLL remains preferable because it gives strong corrective updates to low-probability tokens. This assessment can be guided by knowledge of the pretraining mixture when available, and more directly by measuring predicted probabilities on downstream examples. These observations also suggest that highly specialized models, such as strong mathematical reasoners, may benefit most from high-quality and relevant pretraining: once the prior is strong, lightweight SFT with an appropriate objective can be highly effective. Beyond SFT, prior-leaning objectives may also be useful during late-stage pretraining, where raw web-scale corpora contain substantial noise.

Our findings point to several natural future directions. First, adaptive objectives could adjust their prior-averse or prior-leaning behavior as training progresses, rather than using a fixed objective throughout SFT. Such objectives may be especially useful in the intermediate regime, where neither prior-leaning nor prior-averse objective is uniformly preferred, and the right emphasis may change as the model improves. Second, a more general theoretical treatment could relax the stylized assumptions used in our analysis and characterize a broader range of training dynamics across the continuum. Third, developing better quantitative measures of model capability is an important direction. Predicted probability is a useful and accessible proxy, but it does not always coincide with true capability, especially in settings with miscalibration or confident hallucinations. More refined uncertainty quantification and capability diagnostics could therefore make objective selection more reliable in practice.

## Acknowledgement

We sincerely thank Ziqi Wang for his constructive advice and valuable engagement throughout this project. His sharp suggestions, thoughtful feedback, and sustained discussions substantially shaped and improved this work.

This work is supported by NSF (2134079) and DARPA ITM Program No. FA8650-23-C-7316. The content of the information in this document does not necessarily reflect the position or the policy of the Government, and no official endorsement should be inferred. The U.S. Government is authorized to reproduce and distribute reprints for Government purposes notwithstanding any copyright notation

here on. This research used both the DeltaAI advanced computing and data resource, which is supported by the National Science Foundation (award OAC 2320345) and the State of Illinois, and the Delta advanced computing and data resource which is supported by the National Science Foundation (award OAC 2005572) and the State of Illinois. Delta and DeltaAI are joint efforts of the University of Illinois Urbana-Champaign and its National Center for Supercomputing Applications. This work was supported by allocation CIS250611 from the Advanced Cyberinfrastructure Coordination Ecosystem: Services & Support (ACCESS) program, which is supported by National Science Foundation grants #2138259, #2138286, #2138307, #2137603, and #2138296.

## Impact Statement

This paper presents work whose goal is to advance the field of Machine Learning. There are many potential societal consequences of our work, none of which we feel must be specifically highlighted here.

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

## Contents

## A. Related Works

**Language Model Post-training.** Supervised Fine-Tuning (SFT) has emerged as the dominant paradigm for post-training, adapting pretrained models to tasks or domains by directly fitting labeled data (Zhang et al., 2026b; Chung et al., 2024). The availability of high-quality instruction datasets (Mishra et al., 2022; Zhou et al., 2023; Taori et al., 2023; Lightman et al., 2023) has further boosted SFT's effectiveness. Nevertheless, abundament studies highlight that SFT alone often overfits, generalizes poorly, and yields sub-optimal models (Howard & Ruder, 2018; Dodge et al., 2020; Ouyang et al., 2022). To address these limitations while retaining SFT's efficiency, the prevailing recipe is to combine SFT with RL, forming the de facto post-training paradigm (Bai et al., 2022; Achiam et al., 2023; Kirk et al.; Chu et al., 2025; Liu et al., 2026). Yet, existing SFT post-training consistently minimizes the negative log-likelihood objective, $-\log(p)$, whose suitability has rarely been questioned. In this work, we show that it is not universally optimal and argue for revisiting objectives that better exploit pretrained priors in SFT.

**Improving SFT (from an RL perspective).** Motivated by the success of reinforcement learning in reasoning tasks, a growing body of work seeks to reinterpret and improve SFT through an RL lens. Wang et al. (2026) cast both SFT and DPO as instances of implicit reward learning, showing that smaller learning rates and alternative divergence-based objectives can enhance performance. Qin & Springenberg (2025) integrates importance sampling into SFT, while Zhu et al. (2026) introduces a PPO-style clipped surrogate objective to constrain policy drift. Aghajanyan et al. (2021) introduces an explicit KL regularizer during fine-tuning, which can also be viewed as a prior-leaning mechanism that discourages unnecessary deviation from the pretrained model. Most closely related to our work, Wu et al. (2026) proposes reweighting gradient coefficients uniformly, essentially equivalent to our $-p$ objective, for which we provide a deeper characterization and analysis. Overall, these approaches can be regarded as special cases of our proposed "prior-leaning" objectives, implemented through RL techniques to downweight low-probability tokens. In contrast, we show that the same effect can be achieved far more simply by applying a threshold. Moreover, these RL-inspired methods are only validated in a single domain, whereas we demonstrate the potential limitations of prior-leaning objectives in the model-weak end. Other than RL-inspired approaches, Zhang et al. (2026a) further explore data selection by favoring high-probability instances, a weaker form of our token-wise thresholding objective.

**Classical views on SFT learning objectives.** In the conventional view of classification, the NLL has long been regarded as the optimal training objective: it is the maximum likelihood estimator (statistical consistency) (Cox, 1958; Casella & Berger, 2024), equivalent to minimizing cross-entropy/KL-divergence (information-theoretic) (Cover, 1999), the unique strictly proper local scoring rule ensuring calibrated probabilities (decision-theoretic) (Savage, 1971; Gneiting & Raftery, 2007), and a convex surrogate to 0-1 loss guaranteeing Bayes consistency and tractable optimization (learning-theoretic) (Bartlett et al., 2006; Zhang, 2004). These arguments, however, assume training from scratch on simple classification tasks, whereas SFT in language model post-training starts from powerful pretrained models with long chain-of-thought supervision where only final answers are evaluated and intermediate tokens may be noisy. Under these conditions, the premises for $-\log(p)$ might no longer hold, and in this work, we provide the first systematic characterization of such settings. We provide a detailed discussion with other loss functions in Appen. F.2.

## B. Detailed Experimental Setup

*Table 4.* General experimental setup across different regions of the model-capability continuum.

| Continuum | Domain | Signals | Training Data | Evaluation Data | Objectives to Compare |
|---|---|---|---|---|---|
| MS | math-reasoning | sparse | NuminaMath CoT | Math500, Minerva Math, Olympiad Bench, AIME24, AMC23 | -p, -log(p), threshold(-log(p)) |
| MI | medical-reasoning | sparse | m23k | MedMC, MedQA, PubMed, MMLU-P, GPQA, Lancet, MedB(4), MedB(5), MedX, NEJM | -p, -log(p) |
| MW | text games | dense | synthetic | synthetic | -p, -log(p) |

We now provide details of our experimental setup, including the rationale for the choice of datasets across the continuum, the corresponding training and evaluation benchmarks, and specific training protocols. An overview is summarized in Tab. 4.

*Table 5.* Continuum selection based on mean predicted probability (Eq. 5). In the MS end, base models already achieve high likelihood on the training set before fine-tuning; in the MI region, predictions are around 0.5; in the MW end, predictions are near zero.

| **Model Strong (Math)** | | | |
|---|---|---|---|
| **Mean Predicted Probability** | 0.80 | 0.76 | 0.80 | 0.81 |
| **Model Name** | LLaMA-3.1-8B | DeepSeekMath-7B | Qwen2.5-Math-1.5B | Qwen2.5-Math-7B |
| **Model Intermediate (Med)** | | | |
| **Mean Predicted Probability** | 0.50 | 0.53 | 0.56 | 0.59 |
| **Model Name** | LLaMA-3.2-3B | LLaMA-3.1-8B | Qwen2.5-1.5B | Qwen2.5-Math-7B |
| **Model Weak (Puzzles)** | | | |
| **Mean Predicted Probability** | 0.01 | 0.01 | 0.01 | 0.07 |
| **Model Name** | LLaMA-3.2-3B | LLaMA-3.1-8B | Qwen2.5-1.5B | Qwen2.5-7B |

## B.1. Continuum Selection

We assign math tasks to the MS end, medical tasks to the MI region, and figfont puzzles to the MW end. For the MS end, we use LLaMA-3.1-8B, DeepSeekMath-7B, Qwen2.5-Math-1.5B, and Qwen2.5-Math-7B. For the MI region, we use LLaMA-3.2-3B, LLaMA-3.1-8B, Qwen2.5-1.5B, and Qwen2.5-Math-7B. For the MW end, we use LLaMA-3.2-3B, LLaMA-3.1-8B, Qwen2.5-1.5B, and Qwen2.5-7B. We rely on base models in all cases.

Our rationale for this selection is twofold.

**First, evidence from pretraining corpora.** Fig. 2 illustrates that some domains are strongly represented in pretraining while others are not. For example, open-sourced documentation of LLaMA-3 reports that ∼25% of pretraining tokens are math-related (Grattafiori et al., 2024), indicating strong priors for math reasoning. Similarly, DeepSeekMath and Qwen2.5-Math were explicitly pretrained on math corpora. By contrast, medical corpora are only partially present in pretraining, yielding moderate priors, and figfont puzzles are completely absent, making them a natural MW task.

**Second, quantitative evidence from model predictions.** Tab. 5 shows mean predicted probabilities (Eq. 5) on the training set, which we use as a proxy for prior strength given that base LLMs are generally well-calibrated and their predictions more faithfully reflect inherent model capability (Zhu et al., 2023; Xie et al., 2024) . In the MS end, models already achieve very high likelihoods (around 0.8) before fine-tuning. In the MW end, predictions are close to zero, reflecting a lack of relevant prior knowledge. In between, predictions cluster around 0.5, reflecting an intermediate level of task familiarity. Together, these observations justify our continuum classification and ground it in both qualitative and quantitative evidence.

## B.2. Training and Evaluation Details

**General framework.** All SFT experiments are conducted using `verl` (Sheng et al., 2024). We fix the optimizer to AdamW, with a base learning rate of $5 \times 10^{-5}$ for all models except LLaMA-3.1-8B, where we use $2 \times 10^{-5}$. We employ cosine decay scheduling with a warm-up ratio of 0.1, and train for a single epoch. All training runs are performed on 2 H200 GPUs with a single node. We have initially tuned the learning rate over $\{1e-3, 5e-4, 1e-4, 1e-5, 2e-5, 5e-5, 1e-6, 5e-6\}$ with one single model on each setting and do not observe notable performance changes when the learning rate is around $5e-4$ to $1e-6$ with the objectives we studied in this paper. Therefore, we generally fix the learning rate to ensure a fair and computationally efficient comparison.

**Model-Strong (Math).** Our setup for mathematical reasoning largely follows Wu et al. (2026). We train on NuminaMath-CoT (LI et al., 2024), which contains 859k chain-of-thought problems collected from multiple sources. For efficiency, we sample a 67k subset, which we find to achieve equivalent performance to larger subsets (100k+ or more). We set the maximum training length to 3072 tokens and use a micro-batch size of 4. Evaluation covers five representative math benchmarks: Math500 (Hendrycks et al., 2021), Minerva Math (Lewkowycz et al., 2022), Olympiad Bench (AI Mathematical Olympiad Prize, 2024), AIME24 (Mathematical Association of America, 2024), and AMC23 (Mathematical Association of America, 2023). Each evaluation uses temperature 1.0, with results reported as the average of 16 generations per example and a maximum generation length of 4096 tokens.

**Model-Intermediate (Medical).** We train on m23k (Huang et al., 2026), a 23k-instance medical reasoning dataset. We experimented with two variants: (i) including long-form reasoning traces (maximum length 8192, micro-batch size 1) and (ii) using only standard chain-of-thought (maximum length 1024, micro-batch size 16). Since performance was similar, we report results from the standard CoT variant. Evaluation strictly follows the protocol in Huang et al. (2026), using temperature 0 and random seed 42. Benchmarks include MedMCQA (Pal et al., 2022), MedQA-USMLE (Jin et al., 2021), PubMedQA (Jin et al., 2019), MMLU-Pro (Wang et al., 2024), GPQA (Medical) (Rein et al., 2024), Lancet & NEJM (Huang et al., 2026), MedBullets (Chen et al., 2025), and MedXpertQA (Zuo et al., 2025). A detailed overview of these datasets is provided in Huang et al. (2026).

**Model-Weak (Figfont).** We generate synthetic figfont puzzles from ReasoningGym (Stojanovski et al., 2025). We generate synthetic figfont puzzle data from ReasoningGym (Stojanovski et al., 2025), creating 40k instances for training and 20k for evaluation. An example puzzle is shown in Fig. 2. Training mirrors the MI setup, with a maximum sequence length of 800 and a micro-batch size of 16. Inference uses temperature 0 and random seed 42. We evaluate with two metrics: (i) exact match and (ii) Jaro–Winkler similarity, a string-based similarity score that is more tolerant to small variations and complements the strictness of exact match.

## C. Additional Experiment Results

### C.1. Justification for Assumptions

*Table 6.* Fraction of training-set tokens whose initial predicted probability exceeds 0.55 in the MS end. The high fractions indicate that the base models already assign substantial probability mass to training targets before fine-tuning, supporting Assump. 6.1.

|  | LLaMA-3.1-8B | DeepSeekMath-7B | Qwen2.5-Math-1.5B | Qwen2.5-Math-7B |
|---|---|---|---|---|
| Percentage of tokens with initial predicted probability larger than 0.55 | 72.8% | 76.7% | 80.6% | 81.2% |

Tab. 6 reports this quantity across the MS backbones used in our experiments.

### C.2. General Instruction Tuning Experiments

To demonstrate that our continuum extends beyond specialized domains (e.g., math or medical QA), we additionally consider general conversational alignment, where the goal is to match subtle human preferences and stylistic behaviors. We construct a mixed SFT corpus from two well-known, high-quality public datasets: Magpie-Prob-300K-Filtered (Xu et al., 2025) and EvoInstruct (Xu et al., 2024), which together target complex instruction following abilities. For efficiency, we subsample 70k instances from the Magpie-Prob-300K-Filtered dataset and we retain only instances whose total sequence length is at most 2048 tokens. This yields a combined training set of approximately *140K* examples.

We consider three backbones, Qwen2.5-3B, Qwen2.5-7B, and Qwen2.5-14B, and fine-tune each for one epoch with batch sizes of 8, 4, and 2 and learning rates of 5e-5, 2e-6, and 1e-6, respectively. We then compare models trained with the standard NLL objective $-\log p$ versus the prior-leaning objective $-p$ to probe the model-capability continuum in this more general alignment setting. For evaluation, due to limited API budget, we use AlpacaEval2 (Dubois et al., 2024) and directly compare responses from the $-p$ model versus the $-\log p$ model. In the table below, we report the win rate of the prior-leaning model against the NLL baseline.

*Table 7.* Win rates (%) of the prior-leaning objective $-p$ against the NLL baseline $-\log p$ on AlpacaEval2. We report both length-controlled win rate (LC WR) and standard win rate (WR), measuring the fraction of pairwise comparisons where the $-p$ model is preferred over the $-\log p$ model. As the backbone scales from 3B to 14B parameters, the prior-leaning objective transitions from underperforming to outperforming NLL, consistent with our model-capability continuum.

|  | Qwen2.5-3B | Qwen2.5-7B | Qwen2.5-14B |
|---|---|---|---|
| LC WR | 41.0 | 49.6 | 57.5 |
| WR | 42.0 | 49.0 | 57.1 |

In Table 7, we observe a clear continuum-style pattern as the model becomes stronger. For the smaller Qwen2.5-3B

backbone, the prior-leaning objective $-p$ underperforms NLL, achieving only about $41$–$42\%$ win rate, indicating that aggressively trusting the model's prior is detrimental when the backbone is relatively weak. For the intermediate Qwen2.5-7B model, the win rates are close to $50\%$, suggesting that the two objectives behave similarly in this regime. In contrast, for the larger Qwen2.5-14B backbone, the same prior-leaning objective attains around $57$–$58\%$ win rate over NLL on AlpacaEval2, suggesting that once the model's prior is sufficiently strong, emphasizing the model's prior beliefs becomes beneficial even on broad alignment tasks.

### C.3. Additional Model Strong Experiments

**Additional Model-Strong: Coding Task.** To further illustrate the generality of the model-strong end of our continuum, we examine the coding domain using the Qwen2.5-Coder-7B backbone, which is known to contain substantial coding-related knowledge. For evaluation, we adopt the EvalPlus suite (Liu et al., 2023) and consider four standard benchmarks: HumanEval, HumanEval+, MBPP, and MBPP+ (all reported as pass@1 following Liu et al. (2023)). For training, we use Magicoder-OSS-Instruct-75K (Wei et al., 2024), a high-quality instruction-tuning corpus tailored specifically for coding tasks, and fine-tune the model for one epoch.

Tab. 8 shows that, in this coding-heavy, model-strong setting, the prior-leaning objective $-p$ consistently outperforms NLL across all four benchmarks. Relative to the base model, $-p$ delivers large absolute gains (e.g., from $63.6$ to $77.8$ average pass@1). On MBPP and MBPP+, the $-p$ model nearly matches the performance of Qwen2.5-Coder-7B-Instruct ($83.5$ and $71.7$, respectively), despite using only 75K training examples. This provides additional evidence that in domains where the backbone already encodes rich task-relevant knowledge, moving toward the model-strong end with a prior-leaning objective can yield substantial gains.

*Table 8.* Model-strong experiments on coding tasks with Qwen2.5-Coder-7B on the EvalPlus benchmarks (pass@1). The prior-leaning objective $-p$ improves over the NLL objective $-\log p$ and substantially boosts performance over the base model. On MBPP and MBPP+, the $-p$ model approaches the performance of the much more heavily tuned Qwen2.5-Coder-7B-Instruct variant, despite using only 75K instruction-tuning examples.

| Case | HumanEval | | MBPP | | Avg. |
|---|---|---|---|---|---|
| | *HE* | *HE+* | MBPP | MBPP+ | |
| **Qwen2.5-Coder-7B** | | | | | |
| Base | 61.6 | 53.0 | 76.9 | 62.9 | 63.6 |
| -log(p) | 80.4 | 71.3 | 78.8 | 68.3 | 74.7 |
| -p | **83.5** | **75.6** | **83.1** | **69.0** | **77.8** |

**Performance versus Model Scale in the Model-Strong End.** We also study how the benefit of the prior-leaning objective $-p$ evolves with model scale in the model-strong end. Concretely, we evaluate Qwen2.5 models at four sizes (3B, 7B, 14B, and 32B parameters), training and evaluating them on the same mathematical tasks and configurations as in the main paper. For the larger models (14B and 32B), we use slightly smaller learning rates of $2 \times 10^{-5}$ and $10^{-5}$, respectively, to ensure stable optimization.

Fig. 7 plots the performance of $-p$ versus NLL with both axes in log scale. As model size increases, the performance gap in favor of the prior-leaning objective widens monotonically: the gains are modest at 3B, more pronounced at 7B, and largest at 14B and 32B. This trend is consistent with our continuum view: as models become more capable and their pretrained priors more informative, they can exploit prior-leaning objectives more effectively, leading to larger improvements in the model-strong end.

### C.4. Additional Model Weak Experiments

**Low-resource Language Instruction Tuning.** We study instruction tuning in low-resource language settings, which provide natural domains where existing language models remain very weak (Zhong et al., 2024). For evaluation, we use MMLU-ProX (Xuan et al., 2025), a challenging question answering benchmark adapted from MMLU-Pro that covers a wide range of languages and assesses general knowledge and reasoning abilities. Among these, we identify seven low-resource languages for which corresponding instruction-tuning data exist: Marathi (MR), Telugu (TE), Nepali (NE), Swahili (SW), Wolof (WO), Yoruba (YO), and Zulu (ZU).

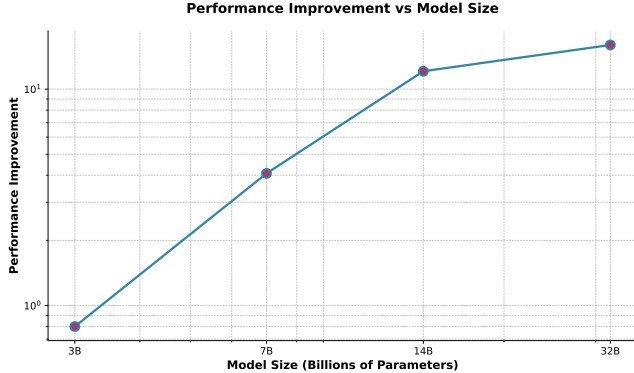

*Figure 7.* Performance improvement of the objective $-p$ over NLL across model scales. Both axes use log scale. Larger models exhibit larger gains, showing that the prior-leaning objective is increasingly effective as the underlying model becomes stronger.

For training, we use the `muri-it` dataset (Köksal et al., 2025), a 2M-example instruction-tuning corpus that spans nearly all languages. We extract the subset corresponding to the seven evaluation languages, resulting in a total of *95K* training instances. We evaluate two backbones, LLaMA-3.1-8B and Qwen2.5-7B, both of which exhibit very limited pretraining exposure to these languages and have extremely low zero-shot performance. Each model is trained for one epoch using the same configuration described earlier in the paper.

In terms of our continuum, these settings are *model-weak*. Tab. 9 presents the results and indeed confirms this. In every low-resource evaluation language and for both backbones, the standard NLL objective substantially outperforms the prior-leaning objective $-p$. This demonstrates that the model-weak end extends far beyond the puzzle cases discussed in the main paper and highlights low-resource multilingual instruction tuning as a natural and practically important instance of this end.

*Table 9.* Additional benchmark results for the model-weak end. Low-resource languages provide a natural setting where pretrained models perform poorly. Best results are in bold.

| Case | Marathi (MR) | Telugu (TE) | Nepali (NE) | Swahili (SW) | Wolof (WO) | Yoruba (YO) | Zulu (ZU) | Avg. |
|------|------|------|------|------|------|------|------|------|
| | | | | **LLaMA-3.1-8B** | | | | |
| Base | 11.1 | 5.0 | 12.3 | 4.1 | 3.7 | 7.4 | 6.0 | 7.1 |
| -p | 15.2 | 2.5 | 15.5 | 14.0 | 11.5 | 10.0 | 11.5 | 11.5 |
| -log(p) | **18.7** | **13.3** | **18.7** | **18.4** | **13.9** | **13.6** | **15.9** | **16.1** |
| | | | | **Qwen2.5-7B** | | | | |
| Base | 8.7 | 2.4 | 9.4 | 8.3 | 9.9 | 7.9 | 7.2 | 7.7 |
| -p | 20.6 | 7.7 | 19.5 | 14.9 | 12.9 | 10.5 | 6.9 | 13.3 |
| -log(p) | **24.8** | **9.1** | **24.4** | **19.9** | **13.1** | **12.6** | **9.6** | **16.2** |

## C.5. Additional Knowledge Memorization Experiments

In this subsection, we demonstrate that our continuum view also applies to a classical knowledge memorization task. We use OpenBookQA (Mihaylov et al., 2018) to study commonsense and factual question answering. The dataset contains 5K training samples and 500 test samples. We fine-tune Qwen2.5-14B and Qwen2.5-7B, both of which achieve over 80% zero-shot accuracy on this benchmark, for one epoch using the same configurations described earlier in the paper.

Importantly, the labels in this setting are few, unambiguous, and essentially noise-free. This places the task in the regime where classical supervised classification applies and where NLL is expected to excel. In contrast, our motivation for prior-leaning objectives comes from the observation that typical reasoning SFT datasets contain imperfect and potentially noisy demonstrations. To emulate the model strong end under such conditions, we perturb the training set with small amounts of label noise and train under the same protocol. This serves as a proxy for *pretraining style noise*, where incorrect or ambiguous labels are common.

Tab. 10 reports the results. In the clean, conventional-classification-like setting, NLL indeed performs best, fully consistent

with our framework. However, once noise is introduced, analogous to imperfections in pretraining corpora and reasoning SFT datasets, the prior-leaning objective $-p$ remains robust while NLL collapses. Moreover, as the model scale increases (14B compared to 7B), the performance gap between prior-leaning and prior-averse objectives in the clean setting becomes smaller. These findings illustrate that the continuum we identify extends beyond long-form reasoning and also governs knowledge memorization under varying supervision quality.

*Table 10.* Additional benchmark results on knowledge memorization (OpenBookQA). The *clean* setting uses the original labels and reflects standard classification with noise-free supervision. The *noisy* setting adds small label perturbations to emulate pretraining-style noise. In the clean setting, NLL performs best, whereas under noisy supervision, the prior-leaning objective $-p$ is substantially more robust, illustrating our continuum beyond reasoning tasks.

| Cases | OpenBookQA | |
|---|---|---|
| | Clean (MW) | Noisy (MS) |
| **Qwen2.5-14B** | | |
| Base | 82.2 | |
| -log(p) | **95.0** | 27.0 |
| -p | 93.4 | **91.6** |
| **Qwen2.5-7B** | | |
| Base | 81.0 | |
| -log(p) | **91.0** | 27.2 |
| -p | 86.2 | **85.0** |

# D. Proofs for Sec. 3

**Lemma D.1** (Gradient Shape). *Let $f : [0,1] \to \mathbb{R}$ be differentiable and nonincreasing. Consider the objective in Eq. , whose step-$t$ contribution depends on the correct-class probability $p_{t,y} = \mathrm{softmax}(z_t)_y$ only through $f(p_{t,y})$. Then the gradient of $\mathcal{L}_f$ with respect to the logits at step $t$ satisfies*

$$\frac{\partial \mathcal{L}_f}{\partial z_{t,i}} = s_f(p_{t,y})\,(\delta_{i,y} - p_{t,i}), \qquad where \quad s_f(p) \triangleq -f'(p)\,p \geq 0.$$

*In particular, for the correct class $i = y$,*

$$\frac{\partial \mathcal{L}_f}{\partial z_{t,y}} = s_f(p_{t,y})\,(1 - p_{t,y}) = W_f(p_{t,y}), \qquad W_f(p) \triangleq -f'(p)\,p\,(1-p).$$

*Proof.* Write $p_t = \mathrm{softmax}(z_t)$, so $p_{t,i} = \exp(z_{t,i})/\sum_j \exp(z_{t,j})$. The softmax Jacobian gives, for all $i$,

$$\frac{\partial p_{t,y}}{\partial z_{t,i}} = p_{t,y}\,(\delta_{i,y} - p_{t,i}).$$

Since the step-$t$ loss is $f(p_{t,y})$, the chain rule yields

$$\frac{\partial \mathcal{L}_f}{\partial z_{t,i}} = f'(p_{t,y})\,\frac{\partial p_{t,y}}{\partial z_{t,i}} = f'(p_{t,y})\,p_{t,y}\,(\delta_{i,y} - p_{t,i}) = \big(-f'(p_{t,y})\,p_{t,y}\big)\,(\delta_{i,y} - p_{t,i}).$$

Define $s_f(p) = -f'(p)\,p$. Because $f$ is nonincreasing, $f'(p) \leq 0$ on $(0,1)$, hence $s_f(p) \geq 0$. The displayed formula then follows, and for $i = y$ we obtain $\frac{\partial \mathcal{L}_f}{\partial z_{t,y}} = s_f(p_{t,y})(1 - p_{t,y}) = -f'(p_{t,y})\,p_{t,y}(1 - p_{t,y}) = W_f(p_{t,y})$. □

**Proposition D.2** (Convex versus Concave Objectives). *Let $f \in C^2[0,1]$ with $f'(p) < 0$ for all $p \in (0,1)$, and define $W_f(p) = -f'(p)\,p(1-p)$. If $f$ is concave ($f'' \leq 0$), then any maximizer of $W_f$ lies in $[\frac{1}{2}, 1]$. If $f$ is convex ($f'' \geq 0$), then any maximizer of $W_f$ lies in $[0, \frac{1}{2}]$.*

*Proof.* Set $s(p) := -f'(p)$. Then $s(p) > 0$ on $(0, 1)$ by the hypothesis $f'(p) < 0$, and

$$W_f(p) = s(p)\, p(1 - p).$$

Differentiate:

$$W_f'(p) = s'(p)\, p(1 - p) + s(p)\, (1 - 2p).$$

*Concave case.* If $f'' \leq 0$ on $[0, 1]$, then $s'(p) = -f''(p) \geq 0$. For $p \in (0, \frac{1}{2})$ we have $1 - 2p > 0$, hence both terms in $W_f'(p)$ are nonnegative; since $s(p) > 0$, in fact $W_f'(p) > 0$ on $(0, \frac{1}{2})$. Therefore $W_f$ is strictly increasing on $(0, \frac{1}{2})$, so no maximizer can lie in $(0, \frac{1}{2})$; any global maximizer must belong to $[\frac{1}{2}, 1]$.

*Convex case.* If $f'' \geq 0$ on $[0, 1]$, then $s'(p) = -f''(p) \leq 0$. For $p \in (\frac{1}{2}, 1)$ we have $1 - 2p < 0$; with $s(p) > 0$ the two terms in $W_f'(p)$ are nonpositive, hence $W_f'(p) < 0$ on $(\frac{1}{2}, 1)$. Thus $W_f$ is strictly decreasing on $(\frac{1}{2}, 1)$, so no maximizer can lie in $(\frac{1}{2}, 1)$; any global maximizer must belong to $[0, \frac{1}{2}]$.

Combining the two cases establishes the claim. □

## E. Main Theoretical Results

### E.1. Setup and Notations

**Data model.** Let the input prompt $x \in \mathcal{X}$. The *true* conditional distribution over tokens $y \in [V]$ is $r(y \mid x)$. We let $\mathcal{D}$ denote the (marginal) distribution over pairs $(x, r(\cdot \mid x))$. We use $T(\cdot \mid x)$ to denote the empirical training distribution over contexts $x$.

**Model and objectives.** Let $q_\theta(\cdot \mid x) = \mathrm{softmax}(z_\theta(x))$ be the next-token distribution of an autoregressive LM with parameters $\theta$, and write $q_0(\cdot \mid x) = q_{\theta_0}(\cdot \mid x)$ for the base model. We note that we use different notations $q$ (instead of $p$) to denote the model predictions in the appendix.

The *population risk* is

$$\mathcal{R}(\theta) \;=\; \mathbb{E}_{(x, y^*) \sim \mathcal{D}, q \sim q_\theta(\cdot \mid x)} \Big[ -\mathbb{1}\{y^* = y^q\} \Big]$$

During SFT we minimize the empirical objective

$$\mathcal{L}_f(\theta) \;=\; \mathbb{E}_{(x, \tilde{y}) \sim T} \big[ f\big(q_\theta(\tilde{y} \mid x)\big) \big]$$

where $f : [0, 1] \to \mathbb{R}$ is differentiable and decreasing.

**Notation.** Let $z_\theta(x) \in \mathbb{R}^V$ denote the pre-softmax logits and $q_\theta(\cdot \mid x) = \mathrm{softmax}(z_\theta(x))$ the next-token distribution. Fix $x$ and suppress its dependence when clear. Define the logit feature map

$$\Phi(x, y) := \nabla_\theta z_{\theta_0}(x, y) \in \mathbb{R}^d, \qquad \Phi(x) := [\Phi(x, 1), \ldots, \Phi(x, V)] \in \mathbb{R}^{d \times V},$$

and its Gram matrix over logits

$$G(x) := \Phi(x)^\top \Phi(x) \in \mathbb{R}^{V \times V}, \qquad G_{y, y'}(x) = \langle \Phi(x, y), \Phi(x, y') \rangle.$$

Write $q := q_{\theta_0}(\cdot \mid x)$, $r := r(\cdot \mid x)$, and $T := T(\cdot \mid x)$. For a differentiable, increasing $f_i : [0, 1] \to \mathbb{R}$, set

$$(\beta_i)_y := T_y\, q_y\, f_i'(q_y), \quad \beta_i \in \mathbb{R}^V, \qquad S_{f_i} := \langle \beta_i, \mathbf{1} \rangle = \sum_{y=1}^V T_y\, q_y\, f_i'(q_y).$$

Define the discrepancy vectors

$$v_* := \big(r^\top q\big)\, q - r \odot q, \qquad v_i := \beta_i - S_{f_i}\, q, \qquad \beta_{12} := \beta_1 - \beta_2,\ S_{12} := S_{f_1} - S_{f_2},\ v_{12} := v_1 - v_2 = \beta_{12} - S_{12} q.$$

Finally, let $g_i := \nabla \mathcal{L}_{f_i}(\theta_0)$, $k_i := \langle \nabla \mathcal{R}(\theta_0), g_i \rangle$ and

$$H_i := \int_0^1 \nabla^2 \mathcal{R}\big(\theta_0 - t\, \eta\, g_i\big)\, dt$$

for a stepsize $\eta > 0$ (used later in second-order expansions).

### E.2. Assumptions

#### E.2.1. MAIN ASSUMPTIONS

**Assumption E.1** (Model-Capability Assumption). We make the following assumptions about data capability in the Model-Strong and Model-Weak ends:

- **Model-Weak.** In the MW end, we assume that model predictions are uniform over the vocabulary $V$, where $2/V < 0.55$.

- **Model-Strong.** In the MS end, we assume that for any given $x$, $\Pr_{y^*, \tilde{y}}\left[(q_{y^*} + q_{\tilde{y}}) \geq 0.55\right] \geq K$ with $K \geq 0.70$.

**Assumption E.2** (Trainable Base Model). We assume that the base model is still not perfect: for any given $x$, $\Pr\left[(0.55 \leq q_{y^*} + q_{\tilde{y}}) \leq 0.95\right] \geq \alpha \Pr_{y^*, \tilde{y}}\left[(q_{y^*} + q_{\tilde{y}}) \leq 0.50\right]$ in the MS end.

These assumptions are mentioned in the main paper with justifications. For a uniform predictor, $q_{y^*} + q_{\tilde{y}} = 2/V$, so $2/V < 0.55$ ensures that the MW assumption does not satisfy the MS threshold. This condition is trivially true for LLM-scale vocabularies. The coefficient $\alpha$ could depend on the task itself, and this value $\geq 1$ in practice. Assumption E.2 is a more general re-statement of Assumption 6.2.

#### E.2.2. ADDITIONAL SIMPLIFICATION ASSUMPTIONS

**Assumption E.3** (Model and Data Simplifications). We assume that the feature matrix $\Phi$ is preconditioned such that all of its singular values are equal to one, and that both the training distribution $T$ and the true distribution $r$ are one-hot.

This assumption is made purely for analytical convenience: it removes irrelevant conditioning factors in the proof and allows us to focus on the essential differences between objectives.

### E.3. Main Proofs

**Lemma E.4** (Gradient identities). *We have the following identities:*

$$\nabla \mathcal{R}(\theta_0) = \mathbb{E}_x\big[\Phi(x)\, v_*(x)\big], \qquad \nabla \mathcal{L}_{f_i}(\theta_0) = \mathbb{E}_x\big[\Phi(x)\, v_i(x)\big],$$

*Proof. Population risk.* With $\mathcal{R}(\theta) = \mathbb{E}_x\big[-r(x)^\top q_\theta(\cdot \mid x)\big]$, for fixed $x$ we have $\partial \mathcal{R}/\partial q = -r$. By the chain rule through softmax,

$$\frac{\partial \mathcal{R}}{\partial z} = J(q)\,(-r) = (q^\top r)\, q - q \odot r,$$

so $\nabla_\theta \mathcal{R}(\theta_0) = \Phi(x)\, \frac{\partial \mathcal{R}}{\partial z} = \Phi(x)\, v_*(x)$. Taking expectation over $x$ yields the first identity.

*General $f_i$-objective.* For $\mathcal{L}_{f_i}(\theta) = \mathbb{E}_x\big[\sum_y T_y(x)\, f_i(q_y)\big]$, $\partial \mathcal{L}_{f_i}/\partial q = m_i$ with $m_i = (T_y f_i'(q_y))_y$. Again, $\partial \mathcal{L}_{f_i}/\partial z = J(q)\, m_i = v_i$, whence $\nabla_\theta \mathcal{L}_{f_i}(\theta_0) = \Phi(x)\, v_i(x)$ and the claim follows after taking expectation over $x$. $\square$

**Lemma E.5** (Functional derivative). *Define*

$$J(f_i) \;:=\; \mathbb{E}_x\left[v_*^\top \Phi^\top \Phi\, v_i - \frac{\eta}{2}\, v_i^\top \Phi^\top H_i\, \Phi\, v_i\right], \quad H_i \;:=\; \int_0^1 \nabla^2 \mathcal{R}\big(\theta_0 - t\,\eta\, g_i\big)\, dt,$$

*with $g_i := \nabla \mathcal{L}_{f_i}(\theta_0) = \mathbb{E}_x[\Phi\, v_i]$, $v_* := (r^\top q)\, q - r \odot q$, $v_i := \beta_i - S_{f_i} q$, $(\beta_i)_y := T_y q_y f_i'(q_y)$, $S_{f_i} = \sum_y T_y q_y f_i'(q_y)$. For a perturbation $h$ of $f_i$ (so that $f_i \mapsto f_i + \epsilon h$), the first variation is*

$$\delta J(f_i; h) = \mathbb{E}_x\big[\big(v_*^\top \Phi^\top \Phi \;-\; \eta\, v_i^\top \Phi^\top H_i \Phi\big)\, \delta v_i\big] \;+\; \frac{\eta^2}{2}\int_0^1 t\, \big\langle \nabla^3 \mathcal{R}\big(\theta_0 - t\eta g_i\big)\,[\delta g_i],\, g_i \otimes g_i \big\rangle\, dt,$$

*where $\delta g_i = \mathbb{E}_x[\Phi\, \delta v_i]$ and*

$$\delta v_i \;=\; \delta \beta_i - (\delta S_{f_i})\, q \;=\; \Big(\mathrm{Diag}(T \odot q) - q\,(T \odot q)^\top\Big)\, h'(q).$$

*Proof.* Write $A := \Phi(x)$ for brevity. Then

$$J = \mathbb{E}_x \left[ v_*^\top A^\top A \, v_i - \frac{\eta}{2} \, v_i^\top A^\top H_i A \, v_i \right].$$

Vary $f_i \mapsto f_i + \epsilon h$. Since $v_*$ is fixed, $\delta(v_*^\top A^\top A v_i) = v_*^\top A^\top A \, \delta v_i$. For the second term, use the product rule:

$$\delta\left(v_i^\top A^\top H_i A \, v_i\right) = 2 \, v_i^\top A^\top H_i A \, \delta v_i \ + \ v_i^\top A^\top (\delta H_i) A \, v_i.$$

Hence

$$\delta J = \mathbb{E}_x \left[ v_*^\top A^\top A \, \delta v_i - \eta \, v_i^\top A^\top H_i A \, \delta v_i - \frac{\eta}{2} \, v_i^\top A^\top (\delta H_i) A \, v_i \right].$$

Now $H_i = \int_0^1 \nabla^2 \mathcal{R}(\theta_0 - t\eta g_i) \, dt$. Since $\delta \nabla^2 \mathcal{R}(\theta) = \nabla^3 \mathcal{R}(\theta)[\,\cdot\,]$ and the evaluation point depends on $g_i$, the chain rule yields

$$\delta H_i = \int_0^1 (-t\eta) \, \nabla^3 \mathcal{R}(\theta_0 - t\eta g_i) \, [\delta g_i] \, dt, \quad \text{with} \quad \delta g_i = \mathbb{E}_x[A \, \delta v_i].$$

Therefore

$$-\frac{\eta}{2} \, v_i^\top A^\top (\delta H_i) A \, v_i = \frac{\eta^2}{2} \int_0^1 t \, \langle \nabla^3 \mathcal{R}(\theta_0 - t\eta g_i) \, [\delta g_i], \, A v_i \otimes A v_i \rangle \, dt.$$

Taking $\mathbb{E}_x$ and using trilinearity in the last two slots, $\mathbb{E}_x \langle \mathcal{T}[\delta g_i], A v_i \otimes A v_i \rangle = \langle \mathcal{T}[\delta g_i], (\mathbb{E}_x A v_i) \otimes (\mathbb{E}_x A v_i) \rangle = \langle \mathcal{T}[\delta g_i], g_i \otimes g_i \rangle$, with $\mathcal{T} := \nabla^3 \mathcal{R}(\cdot)$, gives the stated third-order term.

Finally, the variation of $v_i$ with respect to $f_i$ via $h$ is

$$\delta \beta_i = T \odot q \odot h'(q), \qquad \delta S_{f_i} = \langle T \odot q, \, h'(q) \rangle, \qquad \delta v_i = \delta \beta_i - (\delta S_{f_i}) \, q = \left( \mathrm{Diag}(T \odot q) - q \, (T \odot q)^\top \right) h'(q).$$

Collecting terms yields the claimed formula. $\qquad \square$

**Corollary E.6.** *Define the gradient flow of the following term:*

$$\dot{\mathcal{R}}(\theta_t^{(i)})\big|_{t=0} := \lim_{\eta \searrow 0} \frac{\mathcal{R}(\theta_0) - \mathcal{R}(\theta_1^{(i)})}{\eta} \tag{6}$$

*Then we have*

$$\dot{\mathcal{R}}(\theta_t^{(i)})\big|_{t=0} = \mathbb{E}_x \left[ v_*^\top \Phi^\top \Phi v_i \right] \tag{7}$$

*Proof.* By Taylor Expansion, we have

$$\mathcal{R}(\theta_0) - \mathcal{R}(\theta_1^{(i)}) = \eta \langle \nabla \mathcal{R}(\theta_0), \nabla \mathcal{L}_{f_i}(\theta_0) \rangle - \frac{\eta^2}{2} \nabla \mathcal{L}_{f_i}(\theta_0)^\top \left( \int_0^1 \nabla^2 \mathcal{R}(\theta_0 - t\eta \nabla \mathcal{L}_{f_i}(\theta_0)) \, dt \right) \nabla \mathcal{L}_{f_i}(\theta_0) \tag{8}$$

Then this corollary follows immediately from Lem. E.5. $\qquad \square$

**Lemma E.7** (Useful Inequalities). *Let $q \in \Delta^{V-1}$ be a probability vector and fix an index $j$.*

1. *For all $q$,*
$$q_j^2 \, \|e_j - q\|^2 \ \leq \ 2 q_j^2 (1 - q_j)^2, \tag{9}$$
*and the bound is tight (equality holds) when all mass $\sum_{i \neq j} q_i = 1 - q_j$ is concentrated on a single coordinate.*

2. *For fixed distinct $i \neq j$, consider*
$$F(q) := q_i q_j \left( -q_i - q_j + \|q\|^2 \right).$$

*Then*

$$\max_{q \in \Delta^{V-1}} F(q) = \frac{11\sqrt{33} - 59}{768} \leq 0.00546,$$

*and the maximizer is attained by a vector with*

$$q_i = q_j = \frac{9 - \sqrt{33}}{24}, \qquad \text{all remaining mass } 1 - 2q_i \text{ placed on one coordinate.}$$

3. *If we know* $-q_i - q_j + \|q\|^2 \leq 0$, *then*

$$-q_i - q_j + \|q\|^2 \leq 1 + 2(q_i + q_j)^2 - 3(q_i + q_j)$$

*Proof.* *(1)* Since $q$ is a probability vector with nonnegative coordinates,

$$\|e_j - q\|^2 = (1 - q_j)^2 + \sum_{k \neq j} q_k^2 \leq (1 - q_j)^2 + \left(\sum_{k \neq j} q_k\right)^2 = 2(1 - q_j)^2,$$

because $\sum_{k \neq j} q_k^2 \leq (\sum_{k \neq j} q_k)^2$ for nonnegative terms. Multiplying by $q_j^2$ yields Eq. 9. Equality holds when the entire mass $1 - q_j$ lies on a single coordinate distinct from $j$, in which case $\sum_{k \neq j} q_k^2 = (\sum_{k \neq j} q_k)^2 = (1 - q_j)^2$.

*(2)* Set $a = q_i$, $b = q_j$, and $s = 1 - a - b \geq 0$. Write $\|q\|^2 = a^2 + b^2 + t$ with $t := \sum_{k \neq i,j} q_k^2$. For fixed $a, b$, the objective

$$F(q) = ab\big(-a - b + a^2 + b^2 + t\big)$$

is increasing in $t$ whenever $ab > 0$. Since $t \leq s^2$ with equality iff all the mass $s$ is concentrated on a single coordinate, any maximizer (with $ab > 0$) must satisfy $t = s^2 = (1 - a - b)^2$. Thus we may reduce to the two-variable problem

$$G(a, b) := ab\big(-a - b + a^2 + b^2 + (1 - a - b)^2\big), \qquad a \geq 0, \ b \geq 0, \ a + b \leq 1.$$

It is convenient to reparametrize by

$$u := a + b \in [0, 1], \qquad z := (a - b)^2 \in [0, u^2].$$

Then

$$ab = \frac{u^2 - z}{4}, \qquad a^2 + b^2 = \frac{u^2 + z}{2}, \qquad (1 - a - b)^2 = (1 - u)^2,$$

and a short calculation gives

$$G(u, z) = \frac{1}{4}(u^2 - z)\big(1 - 3u + \tfrac{3}{2}u^2 + \tfrac{z}{2}\big) = \frac{1}{4}(u^2 - z)\big(K(u) + \tfrac{z}{2}\big),$$

where $K(u) := 1 - 3u + \tfrac{3}{2}u^2$.

For each fixed $u$, $G(u, z)$ is a concave quadratic in $z$ (its $z^2$-coefficient is $-\frac{1}{8}$). Hence the $z$-maximizer is

$$z^\star(u) = \min\Big\{\max\{0, \ u^2 - 2K(u)\}, \ u^2\Big\} = \min\Big\{\max\{0, \ -\alpha(u)\}, \ u^2\Big\},$$

where $\alpha(u) := u^2 - 3u + 1$. Equivalently,

$$z^\star(u) = \begin{cases} 0, & \alpha(u) \geq 0 \ (\text{i.e. } u \in \big[0, \frac{3 - \sqrt{5}}{2}\big]), \\ -\alpha(u), & \alpha(u) \leq 0 \text{ and } u \leq \frac{1}{2} \ (\text{i.e. } u \in \big[\frac{3 - \sqrt{5}}{2}, \frac{1}{2}\big]), \\ u^2, & u \geq \frac{1}{2}. \end{cases}$$

Thus:

- If $u \in \big[0, \frac{3 - \sqrt{5}}{2}\big]$, then $z^\star(u) = 0$, so the maximizer over $z$ occurs at $a = b = \frac{u}{2}$ (the symmetric point), and

$$G(u, 0) = \frac{u^2}{4} K(u) = \frac{u^2}{4}\big(1 - 3u + \tfrac{3}{2}u^2\big).$$

- If $u \in \big[\frac{3 - \sqrt{5}}{2}, \frac{1}{2}\big]$, then $z^\star(u) = -\alpha(u)$, and a simplification yields

$$\max_z G(u, z) = G\big(u, z^\star(u)\big) = \frac{(u - 1)^2(2u - 1)^2}{8}.$$

Since $\frac{d}{du}\big[(u - 1)^2(2u - 1)^2/8\big] = \frac{1}{4}(u - 1)(2u - 1)(4u - 3) < 0$ on this interval, the maximum over $u$ here is attained at the left endpoint $u = \frac{3 - \sqrt{5}}{2}$.

- If $u \in [\frac{1}{2}, 1]$, then $z^\star(u) = u^2$, which gives $ab = 0$ and hence $G = 0$.

Therefore the global maximizer must lie in the symmetric regime $z = 0$, i.e., $a = b = x$, with $u = 2x \in [0, \frac{3-\sqrt{5}}{2}]$. In this case

$$G(x) = x^2 (6x^2 - 6x + 1), \qquad x \in \left[0, \tfrac{1}{2}\right].$$

Differentiating,

$$G'(x) = 2x (12x^2 - 9x + 1),$$

so the critical point in $(0, \frac{1}{2})$ satisfies $12x^2 - 9x + 1 = 0$, i.e.

$$x_\star = \frac{9 - \sqrt{33}}{24} \in \left(0, \tfrac{1}{2}\right).$$

Since $G(0) = 0$, $G(\frac{1}{2}) = -\frac{1}{8} < 0$, and $G$ achieves a positive value at $x_\star$, the global maximum is attained at $x_\star$. Substituting and simplifying,

$$\max_{q \in \Delta^{V-1}} F(q) = G(x_\star) = \frac{11\sqrt{33} - 59}{768} \le 0.00546.$$

This value is realized by

$$q_i = q_j = x_\star, \qquad q_\ell = 1 - 2x_\star \text{ for some } \ell \notin \{i, j\}, \qquad q_k = 0 \ (k \notin \{i, j, \ell\}),$$

i.e., the remaining mass is concentrated on a single coordinate, as established at the start.

*(3)* We have that

$$
\begin{aligned}
-q_i - q_j + \|q\|^2 &\le -q_i - q_j + q_i^2 + q_j^2 + (1 - q_i - q_j)^2 \\
&= 1 + 2q_i^2 + 2q_j^2 + 2q_i q_j - 3q_i \\
&\le 1 + 2(q_i + q_j)^2 - 3(q_i + q_j)
\end{aligned}
$$

$\square$

**Theorem E.8** (Characterization via Gradient Flow, Restatement of Thm. 6.4). *Under Assumptions E.1- E.3, suppose that $f_2' - f_1'(\tilde{q})$ is negative for all $\tilde{q}$ and that $q_{\tilde{y}} (f_2' - f_1') (q_{\tilde{y}}) > -c$ for some small positive constant $c > 0$ when $q(\tilde{y}) \in [0, 0.55]$ and $q_{\tilde{y}} (f_2' - f_1') (q_{\tilde{y}}) < -d$ for some small positive constant $d$ when $q(\tilde{y}) \in [0.55, 0.95]$ and that $c < 10d$, with an appropriate choice of label noise ( e.g., when $y^* \ne \tilde{y}$) rate $\mathcal{E}$, then we have the following conclusions:*

- $\dot{\mathcal{R}}(\theta_t^{(1)})|_{t=0} \ge \dot{\mathcal{R}}(\theta_t^{(2)})|_{t=0}$ *in Model Strong End.*

- $\dot{\mathcal{R}}(\theta_t^{(1)})|_{t=0} \le \dot{\mathcal{R}}(\theta_t^{(2)})|_{t=0}$ *in Model Weak End.*

*Proof.* By Assumption. E.3, we first expand the following term:

$$\dot{\mathcal{R}}(\theta_t^{(1)})|_{t=0} - \dot{\mathcal{R}}(\theta_t^{(2)})|_{t=0} = \mathbb{E}_x \left[ v_*^\top (v_1 - v_2) \right] \tag{10}$$

$$= \mathbb{E}_x \left[ \left( (r^\top q) q - r \odot q \right)^\top (v_{12}) \right] \tag{11}$$

Note that

$$v_{12} = \sum_y \left[ T_y q_y \left( f_1' - f_2' \right) (q_y) \right] e_y - \left[ \sum_y \left( T_y q_y \right) \left( f_1' - f_2' \right) (q_y) \right] q \tag{12}$$

$$= q_{\tilde{y}} \left( f_1' - f_2' \right) (q_{\tilde{y}}) e_{\tilde{y}} - q_{\tilde{y}} \left( f_1' - f_2' \right) (q_{\tilde{y}}) q \qquad \text{(Only consider } T \text{ one-hot)}$$

$$= q_{\tilde{y}} \left( f_1' - f_2' \right) (e_{\tilde{y}} - q) \tag{13}$$

We can then proceeed as follows:

$$\dot{\mathcal{R}}(\theta_t^{(1)})\big|_{t=0} - \dot{\mathcal{R}}(\theta_t^{(2)})\big|_{t=0} = \mathbb{E}_x \left[ q_{\tilde{y}} \left( f_2' - f_1' \right) (q_{\tilde{y}}) \langle r \odot q - \left( r^\top q \right) q, e_{\tilde{y}} - q \rangle \right] \tag{14}$$

$$= \mathbb{E}_x \left[ q_{\tilde{y}} \left( f_2' - f_1' \right) (q_{\tilde{y}}) \langle q_{y^*} - q_{y^*} q, e_{\tilde{y}} - q \rangle \right] \qquad (r \text{ is also one-hot})$$

$$= \mathbb{E}_x \left[ q_{\tilde{y}} q_{y^*} \left( f_2' - f_1' \right) (q_{\tilde{y}}) \langle e_{y^*} - q, e_{\tilde{y}} - q \rangle \right] \tag{15}$$

$$= \mathbb{E}_x \left[ q_{\tilde{y}} q_{y^*} \left( f_2' - f_1' \right) (q_{\tilde{y}}) \| e_{y^*} - q \|^2 : \tilde{y} = y^* \right] \tag{16}$$

$$+ \mathbb{E}_x \left[ q_{\tilde{y}} q_{y^*} \left( f_2' - f_1' \right) (q_{\tilde{y}}) \left( -q_{y^*} - q_{\tilde{y}} + \| q \|^2 \right) : \tilde{y} \neq y^* \right] \tag{17}$$

Then we first examine the weak model end, now the model is assumed to output uniform distribution over $V$. Denote the label noise rate to be $\mathcal{E}$. Then we have that

$$\dot{\mathcal{R}}(\theta_t^{(1)})\big|_{t=0} - \dot{\mathcal{R}}(\theta_t^{(2)})\big|_{t=0} = \frac{V-1}{V^3} \left( f_2' - f_1' \right) \left( \frac{1}{V} \right) (1 - \mathcal{E}) \tag{18}$$

$$- \frac{1}{V^3} \left( f_2' - f_1' \right) \left( \frac{1}{V} \right) \mathcal{E} \tag{19}$$

$$= \left( f_2' - f_1' \right) \left( \frac{1}{V} \right) \frac{1}{V^3} \left( (V - 1)(1 - \mathcal{E}) - \mathcal{E} \right) < 0 \tag{20}$$

As long as $\mathcal{E} < \frac{V-1}{V}$ and $\left( f_2' - f_1' \right) \left( \frac{1}{V} \right) < 0$. Then we have the desired condition.

Then we examine strong model end, applying Lemma E.7, we have

$$\mathbb{E}_x \left[ q_{\tilde{y}} q_{y^*} \left( f_2' - f_1' \right) (q_{\tilde{y}}) \| e_{y^*} - q \|^2 : \tilde{y} = y^* \right] \geq 2(1 - \mathcal{E}) \mathbb{E} \left[ \left( f_2' - f_1' \right) (q_{y^*}) q_{y^*}^2 (1 - q_{y^*})^2 \right] \tag{21}$$

and define $R = q_{\tilde{y}} \left( f_2' - f_1' \right) (q_{\tilde{y}})$ and $Q = q_{\tilde{y}} q_{y^*} \left( -q_{y^*} - q_{\tilde{y}} + \| q \|^2 \right)$, then first we show the other term is positive.

$$\frac{1}{\mathcal{E}} \mathbb{E}_x \left[ q_{\tilde{y}} q_{y^*} \left( f_2' - f_1' \right) (q_{\tilde{y}}) \left( -q_{y^*} - q_{\tilde{y}} + \| q \|^2 \right) : \tilde{y} \neq y^* \right] \tag{22}$$

$$= \mathbb{E}_x \left[ QR \right] \tag{23}$$

$$= \mathbb{E}_x \left[ QR : Q \geq 0 \right] + \mathbb{E}_x \left[ QR : Q < 0 \right] \tag{24}$$

$$\geq -c \mathbb{E}_x \left[ Q : Q \geq 0 \right] + \mathbb{E}_x \left[ QR : Q < 0 \right] \tag{25}$$

$$\geq -c \Pr \left[ Q \geq 0 \right] * 0.00546 + \mathbb{E}_x \left[ QR : Q < 0 \right] \tag{26}$$

$$> 0 \tag{27}$$

For the last inequality, we can proceed as follows:

$$\mathbb{E}_x \left[ QR \colon Q < 0 \right] - c \Pr \left[ Q \geq 0 \right] * 0.00546$$

$$\geq d * \Pr_{\tilde{y}, y^*} \left[ 0.95 \geq q_{\tilde{y}} + q_{y^*} \geq 0.55 \right] * \min_{0.95 \geq q_{\tilde{y}} + q_{y^*} \geq 0.55} |Q| - c \Pr \left[ q_{\tilde{y}} + q_{y^*} \leq 0.50 \right] * 0.00546$$

$$= d * \Pr_{\tilde{y}, y^*} \left[ 0.95 \geq q_{\tilde{y}} + q_{y^*} \geq 0.55 \right] * 0.045 - c \Pr \left[ q_{\tilde{y}} + q_{y^*} \leq 0.50 \right] * 0.00546$$

$$> 0$$

where the first inequality comes from the sufficient condition for guaranteeing $Q > 0$ is $\Pr_{\tilde{y}, y^*} \left[ q_{\tilde{y}} + q_{y^*} > 0.50 \right]$, and by (3) in Lem. E.7, we have that given $Q < 0$,

$$\min_{0.95 \geq q_{\tilde{y}} + q_{y^*} \geq 0.55} |Q| \leq - \max_{0.95 \geq q_{\tilde{y}} + q_{y^*} \geq 0.55} 1 + 2(q_{\tilde{y}} + q_{y^*})^2 - 3(q_{\tilde{y}} + q_{y^*}) \leq 0.045$$

Also by Assumpion. 6.1 and 6.2, we have $\Pr_{\tilde{y}, y^*} \left[ 0.95 \geq q_{\tilde{y}} + q_{y^*} \geq 0.55 \right] \geq \alpha \Pr \left[ q_{\tilde{y}} + q_{y^*} \leq 0.50 \right]$. Therefore, we have finished the claim.

Therefore, with an appropriate scale of $\mathcal{E}$, specifically with $\mathcal{E} > \frac{|A|}{B-A}$ where $B = \mathbb{E}_x \left[ q_{\tilde{y}} q_{y^*} \left( f_2' - f_1' \right) (q_{\tilde{y}}) \left( -q_{y^*} - q_{\tilde{y}} + \|q\|^2 \right) \colon \tilde{y} \neq y^* \right] > 0$ and $A = \mathbb{E}_x \left[ q_{\tilde{y}} q_{y^*} \left( f_2' - f_1' \right) (q_{\tilde{y}}) \|e_{y^*} - q\|^2 \colon \tilde{y} = y^* \right] < 0$, then we could achieve the desired result. $\square$

## F. Discussion with Existing Literature

### F.1. Connections with RL

Our analysis is formulated entirely in the supervised SFT setting, where we study probability-based token-level objectives of the form

$$L_f(\theta) = \mathbb{E}_{(x,\tilde{y}) \sim \mathcal{D}} \left[ f \left( p_\theta(\tilde{y} \mid x) \right) \right], \tag{28}$$

on a fixed *offline* dataset $\mathcal{D}$. Here, the training distribution is independent of the current model $\pi_\theta$, and coverage is entirely determined by $\mathcal{D}$.

By contrast, RL methods optimize a sequence-level objective

$$J(\theta) = \mathbb{E}_{x \sim \mathcal{D}, y \sim \pi_\theta(\cdot \mid x)} \left[ r(x, y) \right], \tag{29}$$

where $r(x, y)$ is a scalar reward and $\pi_\theta$ is updated using *online* trajectories sampled from itself. Gradient estimates typically take the form

$$\nabla_\theta J(\theta) = \mathbb{E}_{x \sim \mathcal{D}, y \sim \pi_\theta(\cdot \mid x)} \left[ A(x, y), \nabla_\theta \log \pi_\theta(y \mid x) \right], \tag{30}$$

for some advantage term $A(x, y)$. Because $y$ is drawn from $\pi_\theta$, most gradient mass comes from sequences and tokens that are already high probability under the current policy. This online nature naturally biases updates toward existing high-probability behaviors.

Recent work on RL for LLM reasoning (Davis & Recht, 2025) shows that, for binary correctness rewards, several popular RL-style post-training algorithms can be interpreted as stochastic gradient ascent on a monotone transform of the probability of producing a correct answer given a prompt. If we denote

$$p_\theta^{\mathrm{corr}}(x) := \sum_{y \in \mathcal{Y}^{\mathrm{corr}}(x)} \pi_\theta(y \mid x), \tag{31}$$

then these algorithms optimize an objective of the form

$$J_h(\theta) := \mathbb{E}_{x \sim \mathcal{D}} \left[ h \left( p_\theta^{\mathrm{corr}}(x) \right) \right], \tag{32}$$

for some monotonically increasing function $h(\cdot)$ determined by the algorithm design. From our perspective, Eq. 32 is another instance of the probability-based family in Eq. 28, but applied at the sequence level and coupled to an on-policy sampling scheme.

Practically, RLHF/RLVR pipelines start from a strong base model—typically after extensive pretraining and sometimes a specialized midtraining phase—and include a KL penalty that keeps $\pi_\theta$ close to this base policy (Ouyang et al., 2022; Shao et al., 2024). Combined with the on-policy gradient, this means that updates are dominated by already high-probability sequences, while low-probability ones receive very little gradient signal. This behavior is closely aligned with our *prior-leaning* objectives in the model-strong regime: both favor trusting the pretrained prior when it is reliable. At the same time, RL-based methods come with their own challenges (e.g., exploration versus exploitation, reward misspecification) that are largely orthogonal to the off-policy SFT setting we focus on. A full theoretical unification of RLHF/RLVR with our capability-based continuum is beyond the scope of this work, but Eq. 29–32 highlight that many RL objectives can be naturally interpreted within the same probability-based lens developed here, and extending our framework to fully encompass on-policy RL settings is an exciting direction for future work.

### F.2. Connections with Other Loss Functions

Existing work on alternative SFT losses can be broadly divided into two categories. *Distribution-based* and *Non-Distribution-based* losses. Distribution-based operate directly on the (scalar) probability assigned to the correct label (or a set of correct sequences), and thus fit exactly into our probability-based family $L_f(p)$, while the latter ones are typically composite objectives (e.g., sums of multiple terms, or set/sequence-level surrogates) that depend on the joint behavior of many tokens and do not reduce to a clean function of $p_\theta(\tilde{y} \mid x)$.

**Distribution-based Losses.**

We first discuss *distribution-based* losses, which are the most fundamental and admit a clean characterization through the logit-gradient weight

$$W_f(p) := -f'(p), p(1-p), \tag{33}$$

as we established in Lemma 3.1. As illustrative examples, we analyze the Focal loss by Rege Cambrin et al. (2024) and a Huber-style loss on probabilities, and interpret both within our $W_f(p)$ view.

**Focal loss: a prior-averse example.** Focal loss (Lin et al., 2017; Rege Cambrin et al., 2024) was introduced to address class imbalance by downweighting easy examples and emphasizing hard ones. For a single correct token with probability $p \in (0, 1)$, the (binary) Focal loss can be written as

$$f_{\mathrm{FL}}(p) = -(1-p)^\gamma \log p, \qquad \gamma > 0. \tag{34}$$

A direct calculation yields

$$f'_{\mathrm{FL}}(p) = \frac{(1-p)^{\gamma-1}}{p}\big(\gamma p \log p - (1-p)\big), \tag{35}$$

and therefore

$$W_{f_{\mathrm{FL}}}(p) = -f'_{\mathrm{FL}}(p)p(1-p) = (1-p)^\gamma\big((1-p) - \gamma p \log p\big). \tag{36}$$

Compared to NLL, whose weight is $W_{\mathrm{NLL}}(p) = 1 - p$, Focal loss multiplies this factor by $(1-p)^\gamma$ and introduces the additional term $-\gamma p \log p$. For small $p$ (hard, low-probability tokens), $(1-p)^\gamma$ is close to one and the $-\gamma p \log p$ term is positive and large, so $W_{f_{\mathrm{FL}}}(p)$ can substantially exceed $W_{\mathrm{NLL}}(p)$. For $p$ near one (easy, high-probability tokens), both $(1-p)^\gamma$ and $-\gamma p \log p$ are small, and the weight decays quickly. In our terminology, Focal loss is therefore *more prior-averse* than NLL: it further shifts gradient mass toward low-probability tokens and away from high-probability ones. This behavior aligns with its original motivation of focusing learning on rare or difficult examples, and fits naturally into the model-weak end of our continuum.

**Huber-style loss: a prior-leaning example.** To illustrate a contrasting, more prior-leaning distribution-based objective, we consider a Huber-style loss applied to the probability of the correct token. Let $e = 1 - p$ denote the error in the correct-class probability and $\delta \in (0, 1]$ be a threshold. The Huber loss on $e$ is

$$\phi_\delta(e) = \begin{cases} \frac{1}{2}e^2, & e \leq \delta \\ \delta\big(e - \frac{1}{2}\delta\big) & e > \delta, \end{cases} \tag{37}$$

and we define the probability-based loss

$$f_{\text{Huber}}(p) := \phi_\delta(1 - p). \tag{38}$$

For $e = 1 - p \leq \delta$ (i.e., $p$ close to 1), we have $\phi'_\delta(e) = e$ and thus

$$f'_{\text{Huber}}(p) = -(1 - p), \qquad W_{f_{\text{Huber}}}(p) = -f'_{\text{Huber}}(p) \, p(1 - p) = p(1 - p)^2. \tag{39}$$

For $e = 1 - p > \delta$ (i.e., low-probability, high-error region), $\phi'_\delta(e) = \delta$ is constant and

$$f'_{\text{Huber}}(p) = -\delta, \qquad W_{f_{\text{Huber}}}(p) = \delta \, p(1 - p). \tag{40}$$

Compared to NLL, this Huber-style loss strongly *downweights* both very low- and very high-probability tokens: in the high-confidence region, the weight decays as $p(1 - p)^2$, which is smaller than $1 - p$ for $p$ close to 1; in the low-confidence region, the weight is capped at $\delta p(1 - p)$, which can be much smaller than $1 - p$ when $p$ is small. As a result, gradients are concentrated on moderately confident tokens rather than on extremely low-probability ones. In our framework, this makes the Huber-style loss a *prior-leaning* objective, more conservative than NLL in correcting tokens the model currently deems very unlikely, which aligns with regimes where the pretrained prior is already informative but supervision may be noisy.

**Non-Distribution-based Losses.**

Beyond purely distribution-based objectives, several recent works have proposed losses that depend on *sets of tokens* or on *both* the data and model-generated distributions. These are not of the simple form $f(p_\theta(\tilde{y} \mid x))$ and therefore fall outside the characterization by our paper, but they are still informative for our continuum view.

**Dice and region-based losses.** (Rege Cambrin et al., 2024) transfer semantic-segmentation losses (Dice, Generalized Dice, Lovász, Self-Adjusting Dice) to LLM fine-tuning and combine them with cross-entropy via

$$L_{\text{tot}} = \lambda L_{\text{CE}} + (1 - \lambda)L_{\text{seg}}, \qquad \lambda \in [0, 1],$$

where $L_{\text{CE}}$ is applied to all instruction and answer tokens and $L_{\text{seg}}$ is applied only to answer tokens.[1] For binary Dice, given predicted probabilities $p_i \in [0, 1]$ and labels $y_i \in \{0, 1\}$, the Dice score and loss are

$$\text{DS} = \frac{2 \sum_i p_i y_i}{\sum_i p_i^2 + \sum_i y_i^2}, \qquad L_{\text{Dice}} = 1 - \text{DS}. \tag{41}$$

As noted by both Milletari et al. (2016) and Rege Cambrin et al. (2024), Dice-type losses depend on global set-level quantities (e.g., intersection and union over all tokens). Consequently, the gradient with respect to a single token logit couples all tokens through the numerator and denominator of the Dice score. As a result, isolated misclassified tokens can receive relatively small updates when the overall overlap between predicted and gold token sets is already high. These region-based objectives therefore fall outside our token-wise probability-based characterization via $W_f(p)$: the effective weight on each token cannot be written as a function of its own probability alone, and it is not meaningful to classify them as globally "prior-leaning" or "prior-averse" in our sense.

**Entropic distribution matching (GEM).** (Li et al., 2025) propose GEM, an SFT method based on *entropic distribution matching*. Conceptually, they formulate a reverse-KL objective with entropy regularization

$$\max_f \; \mathbb{E}_x\Big[\mathbb{E}_{y \sim f(\cdot \mid x)} \log p(y \mid x) - \mathbb{E}_{y \sim f(\cdot \mid x)} \log f(y \mid x) + \gamma \, \mathbb{E}_{y \sim f(\cdot \mid x)} \log f(y \mid x)\Big], \tag{42}$$

which is equivalent to minimizing a reverse KL divergence $D_{\text{KL}}(f \| p)$ minus an entropy regularizer (i.e., maximizing entropy). Here $p$ denotes the (unknown) data distribution on sequences and $f$ is the model's generative distribution. Li et al. show that, at the population level, the optimal solution to equation 42 satisfies

$$f^\star(y \mid x) \propto p(y \mid x)^{1/(\gamma+1)}, \tag{43}$$

i.e., a strictly concave power transform of $p$ that flattens peaked distributions and increases entropy. Thus, at the sequence level, GEM behaves as a *strongly prior-leaning* objective: it preserves the modes of $p$ while explicitly avoiding

---

[1]See their Sec. 3.4 and Fig. 2 for the combined loss design.

over-concentrating probability mass on any single sequence, which leads to less overfitting and higher output diversity in practice.

Algorithmically, GEM is implemented via a composite generative loss that contrasts log-probabilities of supervised "real" sequences and model-generated "fake" sequences, with the fake distribution $q$ defined as a softened version of $f$ (via a temperature $\beta$). This composite objective cannot be written as a simple $f(p_\theta(\tilde{y} \mid x))$ on ground-truth tokens, so it lies outside the $W_f(p)$ characterization we develop. Nevertheless, Eq. 43 shows that GEM effectively implements a concave, entropy-increasing transform of the underlying sequence-level probabilities and hence sits naturally on the prior-leaning, model-strong side of our continuum, complementary to the token-level objectives we focus on in this paper.

