# OpenReview forum: "Beyond Log Likelihood: Probability-Based Objectives for Supervised Fine-Tuning across the Model Capability Continuum"
_ICML.cc/2026/Conference — ICML 2026 spotlight_

### Official Review · Reviewer_CVc2 · 2026-03-06

**Soundness:** 3
**Presentation:** 2
**Significance:** 2
**Originality:** 3
**Overall Recommendation:** 4
**Confidence:** 3

**Summary:**

This paper challenges the conventional practice of defaulting to Negative Log-Likelihood (NLL, $-\log p$) in the Supervised Fine-Tuning (SFT) of Large Language Models (LLMs). It proposes the "Model-Capability Continuum" framework, advocating that the optimal loss function should align with the model's prior knowledge of a specific task.

**Compliance With Llm Reviewing Policy:**

Affirmed.

**Final Justification:**

Rebuttal answers the weaknesses.

**Key Questions For Authors:**

Theoretical assumptions are detached from reality: The proofs in Section 6 rely heavily on over-idealized assumptions—such as the model's predictive distribution being perfectly uniform at the "weak" end—which ignore the inherent complexity of representation learning in real-world LLMs.

Failure to control core variables: The experiments do not disentangle "model capability" from "data noise"; thus, the advantage of the $-p$ objective in math tasks likely stems from its robustness to Chain-of-Thought (CoT) noise rather than the model's innate strength.

Circular reasoning in classification: Using "initial predictive probability" as the metric for capability is logically tautological and lacks robustness, as this indicator is highly susceptible to interference from specific prompt formats.

Neglect of high-confidence hallucination risks: Prior-leaning objectives trust the model's prior but lack a safeguard against "confident hallucinations," which may lead to the catastrophic amplification of incorrect but high-probability tokens.

**Limitations:**

The authors must incorporate a dedicated Limitations section, either at the end of the main text or in the appendix, and candidly address the following critical blind spots: The authors should frankly point out that the experimental design has not fully disentangled two confounding factors: the model's innate capabilities and the inherent noise within the datasets (e.g., logical leaps in the Chain-of-Thought data).

**Strengths And Weaknesses:**

1. Theoretical Disconnect
The theoretical derivations in Section 6 are built upon unrealistic "vacuum-spherical" assumptions. For instance, the model's predictive distribution is assumed to be perfectly uniform in the "Model-Weak" end , and the appendix further assumes that the true distribution is one-hot with a feature matrix whose singular values are all equal to one. These strong assumptions strip away the inherent complexity of representation learning in high-dimensional spaces for real-world LLMs, reducing Theorem 6.4 to a mathematical exercise that lacks explanatory power for actual training dynamics.

2. Confounding Variables
The experimental design fails to disentangle the two independent variables: "model capability" and "data noise". In scenarios demonstrating "Model-Strong" capabilities, the paper utilizes mathematical datasets containing complex Chain-of-Thought (CoT) reasoning that are naturally noisy. Conversely, the "Model-Weak" demonstrations rely on pristine, synthetic figfont puzzle data. Consequently, the superior performance of the $-p$ objective on math tasks may likely stem from its inherent robustness to CoT noise rather than the model's purported "innate mathematical strength".

3. Circular Reasoning
The paper employs "mean predicted probability on the training set" as a quantitative proxy for categorizing model strength. Categorizing tasks based on probability and subsequently proving that loss functions sensitive to high probabilities (such as $-p$) perform well on those specific tasks is logically tautological. Furthermore, this probability metric is highly susceptible to interference from different prompt formats and lacks the robustness required for a rigorous benchmark.

4. Missing Baselines & Arbitrary Hyperparameters
Since the authors acknowledge in the related works that these improvements are fundamentally similar to implicit reward learning and Reinforcement Learning (RL) mechanisms , DPO or basic RL must be included as baselines in the main experiments to justify the necessity of purely modifying the SFT loss. Additionally, the $p \ge 0.2$ threshold used in the truncated variants appears arbitrary, lacking support from rigorous grid searches or sensitivity analyses.

---

> ### Author Rebuttal · Authors · 2026-03-26
>
> **W1 & Q1: Thereotical Justifications**
>
> The core contribution of this work is to establish that a broad family of alternative probability-based objectives for SFT is both feasible and meaningfully distinct, and to characterize when these objectives help or hurt. Section 6 is intended as a mechanism analysis, rather than a literal model of full LLM training dynamics. Its assumptions follow a common symmetry-for-tractability pattern in recent high-dimensional and transformer theory. For example, prior work has analyzed multiclass learning with *one-hot labels* and isotropic Gaussian or $N(0, I)$ features [1], transformer ICL with *one-hot labels, uniform class priors*, and isotropic shared covariance [2], and representation-learning settings in the isotropic case $\Sigma=I$, where the optimal decoder has *unit singular values* [3].
>
> In this light, the assumptions in Section 6 should be viewed as tractable abstractions of the two ends of the continuum. The key point is that in this simpler, analyzable regime, these objectives are not equivalent and place qualitatively different emphasis on the training signal. Moreover, our conclusions do not rely on Section 6 alone, but are consistently supported by experiments across 8 backbones, 27 benchmarks, and 7 domains. We will revise the paper to clarify this scope and note that extending the analysis to less restricted settings is an important future direction.
>
> [1]  Theoretical Insights Into Multiclass Classification: A High-dimensional Asymptotic View.
>
> [2] Unlabeled Data Can Provably Enhance In-Context Learning of Transformers.
>
> [3] Fundamental Limits of Two-layer Autoencoders, and Achieving Them with Gradient Methods.
>
> **W2 & Q2: Data Noise and Model Capability**
>
> In our paper, we do not argue that model capability and data noise must be fully disentangled in realistic post-training; rather, stronger pretrained priors should matter more precisely when supervision is noisier.
>
> More importantly, our conclusion is not derived only from “math CoT vs. clean synthetic puzzles.” We show the model-strong end not only in math, but also in coding (App. C.3); the model-intermediate region in medical reasoning (Table 2); and the model-weak end beyond Figfont in a real-world low-resource multilingual instruction-tuning setting (App. C.4). Thus, the continuum we identify is supported across multiple domains and data regimes all with clean CoT traces, rather than by a single comparison. To further address this concern, we additionally audited 150 randomly sampled examples from each dataset with GPT-5.4-mini and found no token-level or sentence-level noise at all, so characterizing these datasets as “naturally noisy” is not supported by our audit.
>
> **W3 & Q3: Probability Metric & Prompt Robustness**
>
> We would like to kindly note that our main claim is about downstream task performance. Mean predicted probability on the training set is used only as a pre-SFT proxy for prior strength when positioning tasks along the continuum; it is neither the optimization target nor the basis of our main conclusion.
>
> If the reviewer is referring to the ablation in Fig. 5, where we compute the mean predicted likelihood on the training set before and after fine-tuning under different objectives, that evidence directly contradicts the alleged “circular reasoning” interpretation. In particular, in the model-weak end, $-log p$ achieves a higher likelihood than $-p$ despite $-p$ directly optimizing probability.
>
> To address prompt sensitivity, we subsampled 1k math examples, rewrote the prompts with GPT-5.4-mini, and re-measured mean predicted probability using LLaMA-3.1-8B. The result is highly stable: 0.79 (original) vs. 0.80 (rewrite).
>
> **W4: Missing Baseline & Thresholded Hyperparameter**
>
> In this paper, our goal is not to propose a universally superior alternative to RL/DPO, but to study SFT itself and demonstrate the existence of a model-capability continuum for probability-based SFT objectives by studying both their success and failure modes. This is mentioned repeatedly in our main paper (e.g., lines 22, 40-42, and 99-103). Because our focus is strictly on the SFT loss landscape, adding DPO or RL baselines, which belong to a separate alignment stage and typically require different data, such as preference pairs, falls outside the scope of this analysis. We believe our experiments across 8 backbones, 27 benchmarks, and 7 domains substantively validate the paper’s SFT-specific claims.
>
> Furthermore, the thresholded objective is mainly an analysis tool rather than our sole methodological proposal. We already conducted a broad sensitivity analysis over thresholds from 0.05 to 0.95, with both bottom and top thresholding, across more than 60 sweeps in Figure 4, Section 5.1.
>
> **Q4: confident hallucinations & Limitation**
>
> Thank you for your sharp observation. Handling confident hallucinations is an important future direction motivated by our work. We will add a limitation and future work section.

---

> > ### Author Rebuttal · Reviewer_CVc2 · 2026-04-04
> >
> > Thanks for your patient response.

---

> > > ### Author Response · Authors · 2026-04-05
> > >
> > > Dear Reviewer CVc2,
> > >
> > > Thank you very much for your positive feedback! We are delighted that our rebuttal has addressed your concerns. We sincerely appreciate your time and effort.
> > >
> > > Best regards,
> > >
> > > Authors of Paper 18994

---

### Official Review · Reviewer_RQWS · 2026-03-13

**Soundness:** 3
**Presentation:** 3
**Significance:** 3
**Originality:** 3
**Overall Recommendation:** 5
**Confidence:** 4

**Summary:**

The paper re-examines whether the standard negative log likelihood (NLL) minimization is the optimal training objective for supervised fine-tuning of LLMs. Specifically, the authors propose a family of probability-based objectives defined by $f(p) = \frac{1-p^\alpha}{\alpha}$ for LLM post-training. The they study the interaction between various $\alpha$ values and model prior strength in LLM post-training by proposing a model-capability continuum. They find that when pretrained models already have strong priors aligned with the task, prior-leaning objectives (like  −p ) work better than NLL. On the other hand, when priors are weak, prior-averse objectives like NLL are preferable. In intermediate regimes, objective choice has minimal / mixed effect. The paper supports this findings with strong empirical results on 8 models and 27 benchmarks as well as provide theoretical insights. In summary the paper provides strong results with theoretical backing. I strongly urge the authors to carefully review the weaknesses and feedback provided to warrant a higher submission score.

**Compliance With Llm Reviewing Policy:**

Affirmed.

**Final Justification:**

Rebuttal answers the weaknesses.

**Key Questions For Authors:**

1. While characterizing which continuum the model/data pair falls into, did the authors experiments with semantic or syntactic similarity measures in-order to avoid running inference over large datasets?
2. There are several methods targeted at robust fine-tuning eg R3F and R4F (https://arxiv.org/abs/2008.03156) which can potentially tackle the prior averse vs favoring regimes. What are the authors thoughts on that?

**Limitations:**

It would help if there is a clear section delineating the limitations of this work (for example, no objective not working for the intermediate stage, etc).

**Strengths And Weaknesses:**

**Strengths:**

1. The paper makes a good algorithmic intervention. While several alternatives to NLL have been proposed, the authors successfully identify that the classical arguments for NLL (maximum likelihood, Bayes consistency, etc) all hold true when training from scratch, but post-training violates these premises. This has been largely understudied before.
2. The experimental scope of the paper is impressive. The authors show promising results 8 models, 27 benchmarks, and 7 domains.
3.  The practical takeaways are clear for what objective to use for prior averse vs prior favoring regimes, albeit being mixed for intermediate setting.
4. The theoretical insights are non-trivial and nicely presented.

**Weakness:**

1. Assumption E.3 with pre-conditioned feature matrix with unit singular values, one-hot training and true distributions, substantially simplifies the analysis. Real LLM training involves correlated features, soft labels, and chain-of-thought sequences where intermediate tokens have complex dependencies.
2.  Despite the framing, the paper largely identifies three discrete regimes rather than a truly continuous characterization. The intermediate region receives the least theoretical treatment where no single objective prevails. A genuine continuum characterization would provide a monotone function relating prior strength (e.g., mean predicted probability) to the optimal $\alpha$ in the parametric family $f(p) = \frac{1-p^\alpha}{\alpha}$. Figure 5 nudges toward this but it is never formalized in the paper.
3. The practical utility of the continuum depends on correctly classifying where a task/model pair sits. The paper uses mean predicted probability as the proxy, but this requires running inference over the training set before training, and the threshold between regimes is not formally specified. Moreover, other stronger uncertainty quantification methods can be a better proxy / combined with NTP logprobs to give a stronger characterization of the continuum.
4. There are no experiments on how the different objectives fare against hyperparameter tuning. For instance, one can increase regularization strength or use low learning rate to avoid overfitting and make NLL work.

---

> ### Author Rebuttal · Authors · 2026-03-26
>
> **W1: Simplified Assumptions & Continuous Characterization**
>
> Our goal is not to reproduce every aspect of full LLM training, but to isolate whether these probability-based objectives are already non-equivalent in a tractable regime. This style of abstraction is common in recent high-dimensional and transformer theory: *one-hot supervision* and *symmetric class priors* [1,2], *isotropic or whitened covariances instead of correlated features* [1,2], later relaxations to structured or anisotropic covariance [3], symmetric spectral structure such as $\Sigma=I$ or *unit-singular-value* decoders [4], and *simplified sequence/token setups* rather than full natural-language chain-of-thought dependencies [5,6]. In this light, Assumption E.3 should be viewed as a mechanism-analysis assumption, not as a literal claim that full-scale LLM training satisfies these conditions exactly. The key point is that in this simpler, analyzable regime, these objectives are not equivalent and place qualitatively different emphasis on the training signal. Our theory focuses on the two ends because this is where the preference reversal is sharpest and the mechanism is analytically cleanest. We will revise the paper to clarify this scope more explicitly, and note that extending the analysis to a fully continuous characterization with relaxed assumptions is an important future direction.
>
> [1] Theoretical Insights Into Multiclass Classification: A High-dimensional Asymptotic View
>
> [2] Unlabeled Data Can Provably Enhance In-Context Learning of Transformers
>
> [3] Gradient-Based Feature Learning under Structured Data
>
> [4] Fundamental Limits of Two-layer Autoencoders, and Achieving Them with Gradient Methods
>
> [5] Context-Parametric Inversion: Why Instruction Finetuning Can Worsen Context Reliance
>
> [6] Transformers Provably Solve Parity Efficiently with Chain of Thought
>
> **W2 & Q1: Continuum Classification and Potential Use of Other Measures**
>
> We intentionally use simple proxies for continuum placement because they are sufficient for the main goal of this paper: to demonstrate the feasibility and existence of a model-capability continuum, and to characterize how a family of probability-based objectives behaves along it. We believe that designing a more precise and automated continuum classifier with exact thresholds, potentially using stronger similarity-based measures, is an exciting direction motivated by our work.
>
> Regarding computation, in practice, one does not need to run inference over the entire training set: using a relatively small subset already gives a similar approximation while substantially reducing cost. For example, we find the same quantitative result on the math dataset with LLaMA-3.1-8B using only a subset of roughly 1k examples.
>
> Regarding uncertainty quantification (UQ), we agree that it is a promising direction for refining continuum characterization. We did not adopt it here because current UQ methods for LLMs are not yet sufficiently stable or consistent to define the continuum cleanly: prior work shows that standard UQ measures can behave unreliably under ambiguity or multiple valid answers, and that different UQ metrics often disagree across prompts and tasks [7,8]. In addition, many strong UQ methods introduce substantial computational overhead, such as repeated sampling or ensembles, making them difficult to apply systematically at the scale of our study. We agree that improved uncertainty-aware measures, semantic/syntactic similarity measures, or their combination with NTP log-probabilities could lead to a more refined characterization of the continuum as the field matures, and we will clarify this in the final version.
>
> [7] The Illusion of Certainty: Uncertainty quantification for LLMs fails under ambiguity.
>
> [8] Look Before You Leap: An Exploratory Study of Uncertainty Analysis for Large Language Models.
>
> **W3: Hyperparameter Sensitivity**
>
> We initially tuned the learning rate and regularization strength with different objectives on one fixed model per domain (details in lines 752-754). Within a reasonable range, we did not observe substantial changes, with performance differences typically below 2%. Since the paper already requires more than 200 training runs, we fixed hyperparameters for the other backbones to keep the study computationally feasible. Overall, our main conclusions remain robust to changes in hyperparameters.
>
> **Q2: Thoughts on some Related Works**
>
> Thank you for this relevant reference. We believe R3F/R4F can be naturally viewed as a prior-leaning intervention, since it explicitly introduces a KL-style constraint toward the original pretrained model. In this sense, the magnitude of the KL term controls the extent to which fine-tuning is encouraged to stay close to the pretrained prior. We will include a discussion of this line of work in the final version.
>
> We will add a limitation section in the final version.

---

> > ### Author Rebuttal · Reviewer_RQWS · 2026-04-03
> >
> > Thanks for the detailed response. Most of my concerns are satisfied.
> >
> > I have updated my scores accordingly.

---

> > > ### Author Response · Authors · 2026-04-03
> > >
> > > Dear Reviewer RQWS,
> > >
> > > Thank you very much for your positive feedback! We are delighted that our rebuttal has addressed your concerns. We sincerely appreciate your time, effort, and support of our work! We have incorporated your suggestions to further strengthen the paper.
> > >
> > > Best regards,
> > >
> > > Authors of Paper 18994

---

### Official Review · Reviewer_LsYb · 2026-03-14

**Soundness:** 3
**Presentation:** 3
**Significance:** 3
**Originality:** 3
**Overall Recommendation:** 5
**Confidence:** 4

**Summary:**

This paper argues that the default supervised fine-tuning objective for large language models, negative log likelihood (NLL), is not uniformly appropriate in post-training, because pretrained models already encode priors and the usefulness of aggressively correcting low-probability tokens depends on how strong those priors are. The authors formalize a family of probability-based token losses, relate their behavior to gradient weighting over token probabilities, and propose a “model-capability continuum” in which prior-leaning objectives such as -p help in model-strong regimes, NLL helps in model-weak regimes, and neither dominates in intermediate regimes. Empirically, they test this claim across math, medical reasoning, and figfont-style puzzle domains, plus additional instruction-tuning analyses, and they provide a stylized gradient-flow theorem intended to explain the reversal in objective preference across strong- and weak-prior settings.

**Compliance With Llm Reviewing Policy:**

Affirmed.

**Final Justification:**

The rebuttal addressed my concerns and improved my evaluation.

**Key Questions For Authors:**

1. Related to the second weakness, would it be possible to disentangle domain effects from prior-strength effects more directly?

2. I'm curious if there are practical implications of this, e.g., guidance for practitioners on what they should use based on some easily measurable properties of their data/task.

3. How do the results change if the training duration or other hyperparameters are altered?

**Limitations:**

Yes.

**Strengths And Weaknesses:**

I enjoyed reading this paper: it offers a clear, testable hypothesis and experiments that back it up, and I think the general insight will be useful for researchers and practitioners.

Some concerns:
1. The novelty claim is a bit overstated -- many recent papers have proposed variants of NLL, and as the authors write, Wu et al. already explored the -p objective etc. I encourage the authors to focus their novelty more explicitly on the capability-dependence that they identify.

2. The continuum is plausible but the experimental design confounds capability with domain. In practice, model-strong = math, intermediate = medical, and model-weak = figfont puzzles. That makes it hard to know whether the effect is really about prior strength, or about other differences across domains.

3. The theory is very stylized. Also, the model-weak assumption (uniform predictions over V) and model-strong assumption (p_y* + p_ỹ ≥ 0.55 with high probability) are only mutually exclusive when V is large. For small V, a uniform model can satisfy both conditions simultaneously, causing the theorem's conclusions to conflict. While this is not a practical issue for LLM-scale vocabularies, the dependence on V should be stated explicitly. More fundamentally, the theory characterizes only two non-overlapping extremes (and only at the initial time point) rather than the continuum that is the paper's central conceptual contribution.

Nit: Table 1's caption really needs to be expanded (or described more in the text). It's somewhat inscrutable.

---

> ### Author Rebuttal · Authors · 2026-03-27
>
> **W1: Positioning of the Work**
>
> We thank the reviewer for highlighting this. In our initial submission, we explicitly emphasized that our primary contribution lies in "capability-dependence" rather than "proposing a single NLL-variant" across multiple sections (e.g., lines 22-31 in the Abstract, lines 39-44 in the Introduction, and lines 99-104 in Related Work with a dedicated "Positioning of our work" subsection). Furthermore, we sought to provide comprehensive coverage of the literature (Sec. 2 and App. A), from classical supervised learning theory to very recent concurrent studies, including the insightful and elegant work by Wu et al. That said, we agree that some broader phrasing may have diluted this focus. In the revision, we will sharpen the wording throughout to make the capability-dependent framing more explicit and consistent.
>
> **W2 & Q1: Domain Effects and Prior Effects**
>
> Thank you for this important question. Our current paper already provides evidence beyond domain differences. First, we study **fixed-domain, varying-capability** settings. In Fig. 3 / App. C.2, on the same general instruction-tuning domain, scaling the backbone from Qwen2.5-3B to 14B shifts -p from underperforming NLL to outperforming it; similarly, in App. C.3 / Fig. 6, under the same math setup, the advantage of -p over NLL grows monotonically with model size. These results directly support a prior-strength interpretation rather than a purely domain-specific one.
>
> Second, the continuum is **not inferred from one domain per regime**. The model-strong end appears not only in math but also in coding (App. C.3), where -p again outperforms NLL; the model-weak end appears not only in Figfont but also in low-resource multilingual instruction tuning (App. C.4), where -log p consistently outperforms -p. Moreover, the same backbone (e.g., LLaMA-3.1-8B) appears across multiple domains in our study. Therefore, while domain-specific factors can certainly exist, the observed reversal is already supported by both **within-domain scaling and cross-domain** replication, rather than by a single, direct math-medical-puzzle comparison.
>
> **W3: Theoretical Assumptions and Implications**
>
> Thank you for your sharp observations and for pointing out the dependence on $|V|$. We will make this technical condition explicit in the final version.
>
> We believe that extending the theoretical result to the entire continuum with more careful analysis of the entire training dynamics is an important and exciting future direction directly inspired by our work. Our central conceptual contribution is consistently supported by experiments across 8 backbones, 27 benchmarks, and 7 domains. The theoretical analysis at Section 6 focuses on the two ends because this is precisely where the preference between objectives reverses most sharply. By contrast, in the intermediate regime, our empirical results show that the gap narrows and no single objective consistently dominates (lines 262-267).  We will revise the paper to clarify this scope more explicitly, and note that extending the analysis to broader settings and the full continuum is an important future direction.
>
> **Q2: Implications for practitioners**
>
> Thank you for this insightful question. We see three practical takeaways.
>
> 1. In many real settings, practitioners know, and often control, the exact composition of pretraining data. This already provides useful intuition about whether a task lies closer to the model-strong or model-weak end. As a more direct proxy, they can also examine the model’s predicted probabilities on a subset of downstream training examples.
> 2. Our results suggest that training a highly specialized model, such as an Olympiad-level math solver, depends primarily on the quality and relevance of pretraining. When the pretrained prior is already strong, post-training through SFT can be lightweight yet effective.
> 3. More broadly, our findings suggest potential value in introducing prior-leaning objectives, directly or gradually, during late-stage pretraining, where learning may otherwise be affected by substantial noise in raw web-scale corpora.
>
> **Q3: Hyperparameter and training duration**
>
> We initially tuned the learning rate and regularization strength with different objectives on one fixed model per domain (details in lines 752–754). Within a reasonable range, we did not observe substantial changes, with performance differences typically below 2%. We also found the results to be stable across training duration: increasing math training examples from 67k to 100k and even 300k yielded almost identical trends (the performance saturates after a certain point). Since the paper requires more than 200 training runs, we fixed hyperparameters for the other backbones to keep the study computationally feasible. Overall, our main conclusions remain robust to changes in hyperparameters and training length.
>
> **Caption Issues**
>
> Thank you for pointing this out. We will expand it for clearer interpretation.

---

> > ### Author Rebuttal · Reviewer_LsYb · 2026-04-03
> >
> > Thanks to the authors for the response. I've updated my score accordingly.

---

> > > ### Author Response · Authors · 2026-04-03
> > >
> > > Dear Reviewer LsYb,
> > >
> > > Thank you very much for your positive feedback! We are delighted that our rebuttal has addressed your concerns. We sincerely appreciate your insightful comments and have incorporated your suggestions to further strengthen the paper.
> > >
> > > Best regards,
> > >
> > > Authors of Paper 18994

---

### Decision · Program_Chairs · 2026-04-30

**Decision:**

Accept (spotlight)

**Comment:**

This paper performs a methodical study of whether modified probability objectives that replace cross-entropy during supervised fine-tuning could help address its shortcomings. They discover that what alternative to choose depends on the model's capabilities, but that a particular low-probability down-weighing approach can provide gains across that range.

The paper is perceived positively by the reviewers. The experiments are exhaustive and paint an insightful picture. Perhaps a deeper theoretical understanding of what explains these observations would make the paper stronger, but the experimental results are worthwhile to share with the community. Readers could benefit from (a) an improved positioning of the paper (e.g., vs. prior uses of $-p$, not competing with RL, etc.), emphasizing that the key contribution is discovering how the choice of alternative changes with model capability, (b) untangling domain effects from model capabilities, (c) more discussion on how the choice of alternative could be made in practice, (d) rule out that hyperparameter tuning with cross-entropy could be enough. Some other adjustments mentioned by the reviewers (e.g., dependence on vocabulary size, captioning figures/tables and referring to them, etc.) should also be addressed. The authors have committed to these changes and are urged to follow through.

The paper brings substantial new insight to supervised fine-tuning and, provided the review feedback is integrated, it is likely to be appreciated by the community.